# Intracellular growth of *Mycobacterium tuberculosis* after macrophage cell death leads to serial killing of host cells

Deeqa Mahamed[1,2], Mikael Boulle[1,2,3], Yashica Ganga[1], Chanelle Mc Arthur[1,2], Steven Skroch[1,2], Lance Oom[1,2], Oana Catinas[1], Kelly Pillay[1,2], Myshnee Naicker[1], Sanisha Rampersad[1,2], Colisile Mathonsi[1,2], Jessica Hunter[1,2], Emily B Wong[1,4], Moosa Suleman[5,6], Gopalkrishna Sreejit[1], Alexander S Pym[1], Gila Lustig[1], Alex Sigal[1,2,3]*

[1]KwaZulu-Natal Research Institute for TB-HIV, Durban, South Africa; [2]University of KwaZulu-Natal, Durban, South Africa; [3]Max Planck Institute for Infection Biology, Berlin, Germany; [4]Division of Infectious Diseases, Massachusetts General Hospital, Boston, United States; [5]Department of Pulmonology and Critical Care, Nelson R Mandela School of Medicine, University of KwaZulu-Natal, Durban, South Africa; [6]Department of Pulmonology, Inkosi Albert Luthuli Central Hospital, Durban, South Africa

**Abstract** A hallmark of pulmonary tuberculosis is the formation of macrophage-rich granulomas. These may restrict *Mycobacterium tuberculosis* (Mtb) growth, or progress to central necrosis and cavitation, facilitating pathogen growth. To determine factors leading to Mtb proliferation and host cell death, we used live cell imaging to track Mtb infection outcomes in individual primary human macrophages. Internalization of Mtb aggregates caused macrophage death, and phagocytosis of large aggregates was more cytotoxic than multiple small aggregates containing similar numbers of bacilli. Macrophage death did not result in clearance of Mtb. Rather, it led to accelerated intracellular Mtb growth regardless of prior activation or macrophage type. In contrast, bacillary replication was controlled in live phagocytes. Mtb grew as a clump in dead cells, and macrophages which internalized dead infected cells were very likely to die themselves, leading to a cell death cascade. This demonstrates how pathogen virulence can be achieved through numbers and aggregation states.

*For correspondence: alex.sigal@k-rith.org

**Competing interests:** The authors declare that no competing interests exist.

## Introduction

Tuberculosis is characterized by the formation of granulomas, cellular structures which attempt to 'wall off' infection by surrounding it with cells of the immune system (*Ramakrishnan, 2012*; *Russell, 2007*; *Russell et al., 2010*). Granulomas, which have a defined anatomical structure as well as segregated expression of immune system related proteins within the structure (*Marakalala et al., 2016*), differentiate and mature independent of each other in infected tissues, most often the lung. The process of pulmonary granuloma formation is driven by macrophage phagocytosis of inhaled, viable Mtb, followed by extravasation of monocytes and T cells from the circulation and their accumulation at the site of infection (*Ramakrishnan, 2012*; *Russell, 2007*; *Russell et al., 2010*; *Barry et al., 2009*). However, granulomas do not always succeed in containing Mtb infection, and different granulomas in the same lung can control the infection or support the growth of the bacilli (*Barry et al., 2009*; *Lenaerts et al., 2015*; *Lin et al., 2014*; *Kaplan et al., 2003*). In the latter case, central necrosis within the granuloma, cavity formation, and breach into the airways, results in

**eLife digest** Every year, around two million people worldwide die from tuberculosis, a disease caused by the bacterium *Mycobacterium tuberculosis* (Mtb). The bacteria generally infect the lungs. In response, the immune system forms structures called granulomas that attempt to control and isolate the infecting pathogens.

Granulomas consist of immune cells known as macrophages, which engulf the *M. tuberculosis* bacteria and isolate them in a cellular compartment where the bacteria either cannot grow or are killed. However, if a large number of macrophages in a granuloma die, the granuloma's core liquefies and the structure is coughed up into the airways, from where *M. tuberculosis* bacteria are transmitted to other people. But how do the bacteria manage to cause the extensive death of the cells that are supposed to control the infection?

By imaging *M. tuberculosis* in human macrophages using time-lapse microscopy, Mahamed et al. reveal that the bacteria break down macrophage control by serially killing macrophages. *M. tuberculosis* cells first clump together and 'gang up' on a macrophage, which engulfs the clump and dies because the bacteria overwhelm it. This does not kill the bacteria, and they rapidly grow inside the dead macrophage. The dead cell is then cleaned up by another macrophage. However, the increasing number of bacteria inside the dead macrophage means that the new macrophage is even more likely to die than the first one. Hence, the bacteria use dead macrophages as fuel to grow on and as bait to attract the next immune cell.

Overall, Mahamed et al. show that once a clump of *M. tuberculosis* initiates death of a single macrophage, it may lead to serial killing of other macrophages and a loss of control over the infection. An important next step will be to understand how the initial clump of bacteria is allowed to form.

---

release of Mtb into the environment and transmission of the infection to another human host (*Ramakrishnan, 2012*; *Russell, 2007*; *Russell et al., 2010*; *Barry et al., 2009*). Thus, while Mtb is considered an intracellular pathogen infecting live cells, its proliferation during active pulmonary tuberculosis occurs in an environment containing many dead cells and cell remnants (*Kaplan et al., 2003*; *Hunter, 2011*; *Hunter et al., 2007*; *Welsh et al., 2011*; *Irwin et al., 2015*).

Part of the host response to Mtb infection is the death of the infected phagocyte. Mtb-induced macrophage death has been observed to occur by apoptotic and non-apoptotic mechanisms. Apoptosis was reported to be protective against Mtb (*Fratazzi et al., 1999*; *Gan et al., 2008*; *Keane et al., 2000*; *Molloy et al., 1994*; *Oddo et al., 1998*; *Behar et al., 2010*). Evidence from mouse macrophages shows that protection is mediated through efferocytosis, the internalization of apoptotic cell fragments by macrophages, leading to the elimination of the Mtb inside the cellular fragments (*Martin et al., 2012*). Mtb was also observed to induce non-apoptotic cell death, differentiated from apoptosis by the lack of caspase activation (*Lee et al., 2006*), lack of DNA fragmentation (*Keane et al., 2000*; *Park et al., 2006*), and loss of membrane integrity (*Park et al., 2006*; *Dobos et al., 2000*). Mtb was not killed when necrotic cell death was induced by $H_2O_2$ (*Molloy et al., 1994*). Mtb is reported to initiate non-apoptotic cell death by the breakdown of mitochondrial membrane integrity (*Duan et al., 2002*), interference with host plasma membrane repair through inhibition of prostaglandin E2 (*Divangahi et al., 2009*; *Divangahi et al., 2010*), and secretion of a necrosis inducing toxin which kills macrophages by hydrolyzing NAD (*Sun et al., 2015*). The inflammatory effect of non-apoptotic cell death resulting from the immediate loss of membrane integrity upon death (*Fink and Cookson, 2005*) may allow for the recruitment of additional host phagocytes to the site of infection which may lead to infection spread (*Clay et al., 2007*). In contrast, control of Mtb through apoptosis and autophagy (*Castillo et al., 2012*; *Gutierrez et al., 2004*; *Watson et al., 2012*) may dampen the inflammatory response.

Mtb infection is known to be controlled by the adaptive immune response (*Cooper and Flynn, 1995*; *Flynn et al., 1995*, *1992*) which becomes effective several weeks after exposure (*Wolf et al., 2008*), though a study using intravital imaging observed limited T cell effector function in Mtb granulomas (*Egen et al., 2011*). The adaptive immune response controls Mtb by activating macrophages

with factors such as interferon gamma (IFNγ) to kill or control the growth of the bacilli, and by targeting infected macrophages with cytotoxic lymphocytes (*Cooper and Flynn, 1995*; *Flynn et al., 1995*, *1992*). However, given that different infection outcomes can coexist in the same lung, the source for variability (*Russell et al., 2009*) in infection outcomes at different foci is not well understood. Substantial variability in other biological systems results from positive feedback (*Kaern et al., 2005*; *Sigal et al., 2006*). Positive feedback involves a cascade of events such that once an event occurs, the probability that it will occur again is increased. To understand how an environment conducive to bacterial growth and host cell death is sustained, we investigated virulent Mtb infection dynamics in primary human macrophages in vitro using time-lapse microscopy over approximately 80 hr at 10 min resolution.

We observed that macrophage internalization of Mtb aggregates led to the killing of the host cell, with the frequency of host cell death increasing with aggregate size. Uptake of single large aggregates showed a higher frequency of cell death relative to the uptake of a similar number of bacteria as multiple small aggregates, and the time to death was inversely correlated to aggregate size. Once the host cell was killed, Mtb was able to rapidly grow inside the dead infected macrophage regardless of whether the cells were monocyte derived macrophages (MDM) or alveolar macrophages, or whether the cells were exposed to IFNγ prior to infection. Growth inside dead cells was significantly faster than in the extracellular environment. Once host cell death occurred, other macrophages internalized the dead infected cells, and this rapidly led to their own death, forming a cell death cascade. These observations of a cycle of host cell death followed by bacterial replication could drive Mtb infection dynamics of necrosis and bacterial proliferation in lung granulomas of patients with active tuberculosis disease.

## Results

### The number and aggregation state of Mtb internalized by individual cells determines macrophage fate

We infected human MDM with the virulent *M. tuberculosis* H37Rv strain labelled with red fluorescent protein (RFP) or mCherry expressed under the control of a constitutively active promoter (groEL and smyc' respectively). We used live cell imaging in Biosafety Level 3 containment over a period of approximately 80 hr to follow the outcome of infection in individual macrophages. MDM cell borders were determined by custom written image analysis software (*Video 1*, Materials and methods, *Source code 1*). Time of cell death was determined to be the time point at which movement of internal cellular structures ceased or the cell detached (Materials and methods). This method of death determination was validated in macrophages with the DNA intercalating dye DRAQ7, which only enters the cell when membrane integrity has been compromised due to cell death (*Figure 1—figure supplement 1*). To examine whether Mtb fluorescence is a valid measure of Mtb number, we tracked the increase in Mtb by fluorescence versus colony forming units (CFU) over 3 days of growth. We found a tight correspondence between the two measures (*Figure 1—figure supplement 2*), with an incremental increase in fluorescence translating to an incremental increase in the number of bacilli as measured by

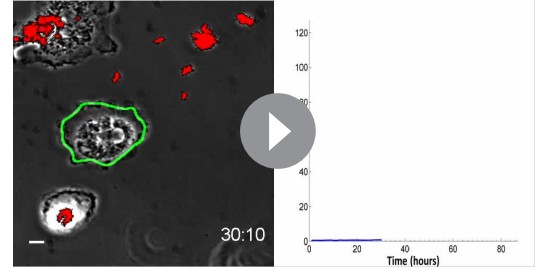

**Video 1.** Macrophage internalization of Mtb aggregates. Mtb infections of human macrophages were imaged by time-lapse microscopy at a resolution of 10 min between image acquisitions. Macrophage borders and locations were tracked using a custom-written, Matlab based image analysis code (*Source code 1*). The left panel of the montage is a movie of macrophage infection. The border of the analyzed macrophage is shown as a green outline. RFP-expressing Mtb are shown in red. Mtb internalized by the cell is marked by blue ellipses. Time is hours: minutes. Scale bar is 20 μm. The graph on right shows the fluorescence signal converted to bacterial numbers in the analyzed macrophage. Timing of internalizations of Mtb clumps containing varying numbers of bacilli as detected from the movie are shown as vertical red lines on the graph. Macrophage death is marked by a vertical black line of x's appearing at the point of cell death.

CFU. Hence, fluorescence measurements reflect Mtb numbers and an increase in Mtb fluorescence reflects Mtb growth.

We asked whether the number of Mtb internalized by MDM, as observed in our time-lapse movies, affected the probability of macrophage death. The aggregation state of Mtb has been hypothesized to impact host-pathogen interactions (*Orme, 2014*; *Sani et al., 2010*). Mtb naturally aggregate, unless cultured with a nonionic detergent such as Tween 80 (*Dubos and Middlebrook, 1948*). To minimize modification of the outer surface of Mtb and its aggregation state (*Sani et al., 2010*), we grew Mtb without detergent for several replication cycles and gently broke up aggregates resulting from such growth into a heterogeneous population of bacterial aggregation states. Hence, MDM in a single experiment were infected by individual bacteria or multibacterial clumps of various sizes (*Video 1*). This enabled us to compare the consequences of MDM phagocytosis of different Mtb numbers and aggregation states in the same experiment. We converted Mtb fluorescent signal inside MDM to bacterial numbers by dividing the signal by the mean fluorescence of a single bacterium, with single Mtb obtained by filtering. We then ranked MDM from lowest to highest by the sum of Mtb internalized. This ranking gave the best separation in the frequency of cell death between the top 50% and bottom 50% of ranks (*Figure 1—figure supplement 3*). The raw data describing the outcome of the macrophage-Mtb interactions is presented in *Figure 1—figure supplement 4*. The full dataset contains the dynamics of macrophage Mtb internalization and fate of 759 MDM, with 720 infected cells and 39 uninfected bystander cells. (*Figure 1—figure supplement 4B*). To compare the death frequencies between cells internalizing different numbers of Mtb over the course of the imaging, we divided the dataset into ten groups of infected cells, where cells internalizing a similar number of Mtb were grouped together and where each group constituted 10% of the total number of infected cells. We also compared death frequency to a group of bystander cells which did not phagocytose any Mtb. We then determined the frequency of cell death in each group, and as well as the mean number of Mtb in the group. Macrophage fate was found to depend on the number of Mtb internalized. As the number of bacteria internalized increased, there was more cell death and it occurred sooner (*Figure 1*). Phagocytosis of small numbers of Mtb, less than approximately 10 bacilli, did not induce significantly more macrophage death than observed in uninfected bystander cells (*Figure 1—figure supplement 5*). Above 10 Mtb internalized but below approximately 30, there was a trend toward increased macrophage death relative to bystander cells, but the differences in frequencies were not significant (see *Figure 1—figure supplement 5B* for p-values and significance thresholds). The probability of death increased significantly relative to bystanders and cells with less than 10 Mtb when more than 30 Mtb were internalized per cell over the course of the movie, and a threshold for Mtb to induce death was clearly visible in *Figure 1* at about 50 Mtb internalized, with differences relative to bystander and lightly infected cells becoming highly significant (*Figure 1—figure supplement 5B*).

We examined whether the frequency of cell death was different if a cell internalized a given amount of Mtb in one pickup as opposed to over multiple pickups. In order to control for bacterial number, we limited the comparison to cells which internalized an average of approximately 50 Mtb (49.6 ± 18.9 Mtb for macrophages which phagocytosed multiple smaller aggregates, versus 49.4 ± 14.3 for macrophages which phagocytosed one large aggregate), an infection level where approximately one half of infected cells died by the end of our imaging period of 83 hr (*Figure 1—figure supplement 5A*). This infection level was chosen so as to increase sensitivity by excluding very highly infected cells, which died with a high frequency, as well as lightly infected cells, which did not die at an appreciably higher frequency relative to bystanders. We obtained a lower frequency of cell death when infection occurred by multiple smaller pickups compared to single large pickups (*Figure 2*). While 47% of the cells died when they accumulated Mtb in three or more internalizations, 71% of the cells died when they internalized a similar mean number of Mtb as one pickup. The difference was significant (p=$5\times10^{-4}$ by bootstrap). In addition to the summary of the results (*Figure 2A*), we also represent the individual cell histories (*Figure 2B* for multiple pickups and *Figure 2C* for single pickups). The data are inherently complex as individual macrophages are tracked, and these phagocytose different numbers of Mtb in different states at different times, and may or may not die as a result. We represent each cell as a line, with pickup events (denoted as circles or stars, where are circles are internalizations of cell-free Mtb, and stars are internalizations of dead infected cells, both color coded according to the number of Mtb internalized) at a location on the line

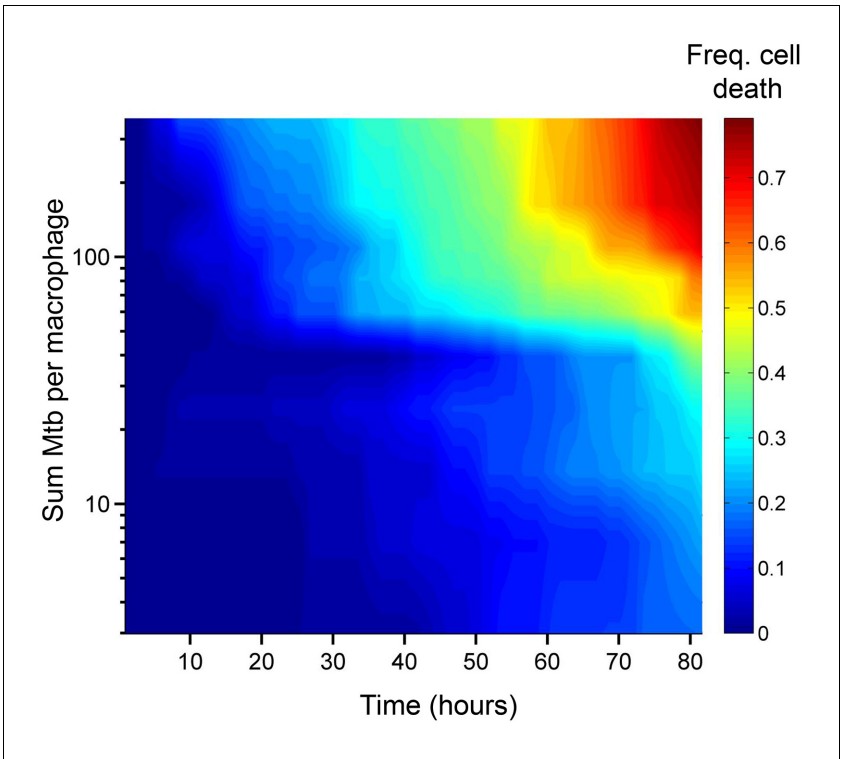

**Figure 1.** Number of Mtb internalized determines probability of macrophage death. The death frequency for 720 infected MDM was graphed over time as a function of the sum of Mtb internalized (raw data provided in *Figure 1—source data 1*). To obtain death frequencies, MDM were divided into 10 groups according to the number of Mtb internalized. The frequency of dead cells was determined at 10 movie frame (1.7 hr) intervals for each group (x-axis), and plotted against the mean sum of Mtb internalized for the group (y-axis). Interpolation between points was performed to obtain a surface. The color scale represents the frequency of cell death from low (blue) to high (red).

The following source data and figure supplements are available for figure 1:

**Source data 1.** Intracellular Mtb fluorescence through time, Movie frames of Mtb phagocytosis, and MDM frame of death, if it occurs, for IFNγ untreated MDM.

**Figure supplement 1.** Measures of cell death.

**Figure supplement 2.** Fluorescence as a measure of Mtb numbers.

**Figure supplement 3.** MDMs were ranked according to the sum of Mtb internalized (black), number of Mtb in the last aggregate internalized (red), number of Mtb in the first aggregate internalized (green), or randomly (blue).

**Figure supplement 4.** Outcomes of macrophage-Mtb interactions.

**Figure supplement 5.** Analysis of differences between macrophages grouped by the sum of Mtb internalized.

corresponding to the time of pickup. If macrophage death occurs, line color changes from green to dark blue at the time of death.

We asked whether time to cell death was dependent on pickup size. We therefore compared macrophages which internalized single clumps of Mtb during the first half of the imaging period, to increase the measurable time to death, if it occurred. In lightly infected cells (≤10 Mtb internalized as one aggregate) cell death was intermittent and not clearly linked to the internalization event. In contrast, the time to cell death appeared to be clearly linked to the time of pickup when the number

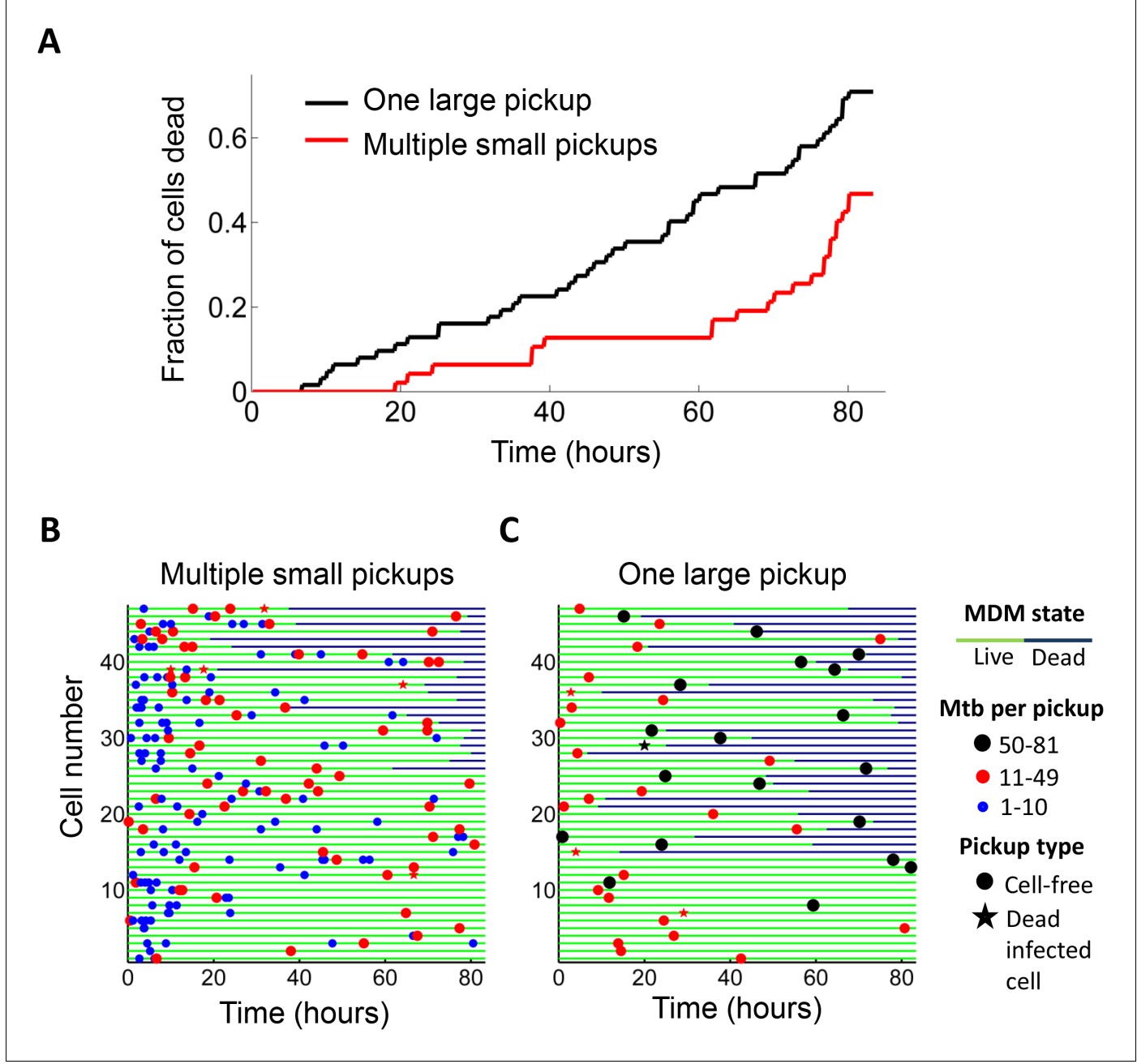

**Figure 2.** Internalization of single large aggregates is more cytotoxic than several smaller aggregates. (**A**) Fraction of dead MDM after phagocytosing one large aggregate of Mtb (black line) or multiple small aggregates (red line) with a similar cumulative sum of Mtb to the single internalizations, but with all aggregates being smaller than 50 bacilli each. The frequency of cell death was 47% for multiple internalizations (n = 47) and 71% for single internalizations (n = 62, p=5×10$^{-4}$ by bootstrap). Individual cell fates are shown in (**B**) for cells internalizing multiple small aggregates, and (**C**) for cells internalizing a similar number of Mtb as single aggregates. The blue circles are internalizations of aggregates of 10 or fewer Mtb, red circles are clumps of 11–49 Mtb, and black circles are 50–81 Mtb. Line color changes from green to dark blue at time of death. Stars indicate internalizations of dead infected cells. An equal number of cells is shown in (**B**) and (**C**) to facilitate comparison. Mean number of Mtb phagocytosed was 49.6 ± 18.9 for MDM internalizing multiple small Mtb clumps, and 49.4 ± 14.3 for MDM internalizing a single large clump.

of Mtb in the clump was large (*Figure 3A*). The time to cell death after internalization inversely correlated with the number of bacteria internalized in a single pickup (*Figure 3B*, $R^2$ = 0.45, p=$2\times10^{-6}$), indicating that uptake of larger clumps led to faster onset of macrophage death.

It has been previously reported that high multiplicity infection with Mtb leads to non-apoptotic cell death (*Molloy et al., 1994*; *Lee et al., 2006*; *Park et al., 2006*; *Dobos et al., 2000*; *Repasy et al., 2013*). To examine the mode of Mtb aggregate-induced cell death, we compared Mtb-induced cell death to that triggered by cisplatin, a chemotherapeutic agent that induces macrophage apoptosis by DNA damage (*von Knethen et al., 1998*), and the toxin nigericin, reported to induce macrophage cell death by pyroptosis following LPS priming (*Warny and Kelly, 1999*). In apoptosis, loss of membrane integrity is delayed to provide an opportunity for the removal of the dead cell (*Fadok et al., 1992*; *Vermes et al., 1995*). In pyroptosis, necrosis and other inflammatory cell death types, cell death involves the immediate loss of membrane integrity and spillage of cell contents (*Fink and Cookson, 2005*). We used DRAQ7 to track loss of membrane integrity, and cessation of internal movement to define the point of cell death (*Figure 4A–B*, Materials and methods). Mtb typically induced phagocyte cell death with immediate loss of membrane integrity (*Video 2*, *Figure 4*), similar to nigericin induced cell death (*Video 3*, *Figure 4*), and consistent with a previous study using A549 epithelial cells (*Dobos et al., 2000*). In contrast, cisplatin induced cell death was apparent significantly before incorporation of DRAQ7 (*Video 4*, *Figure 4*).

## Mtb clumps are not eliminated by macrophage death and rapidly grow inside the dead cell

We tracked the fate of Mtb in the dead cells. Mtb-induced macrophage death did not lead to the killing of the bacilli, and Mtb fluorescence levels increased in the dead cells (*Figure 5A*, *Video 5*). While RFP or mCherry fluorescence possibly persists after Mtb death, the increase in fluorescent signal is a clear indication of Mtb replication (*Figure 1—figure supplement 2*). *Figure 5A* is typical of the type of bacterial growth we observed: Mtb did not burst out of the cell to spread into the neighboring medium. Instead, the bacilli remained tightly associated with the dead host cell as a growing clump. Out of the 92 dead infected cells we evaluated, 90 showed positive growth rates while two

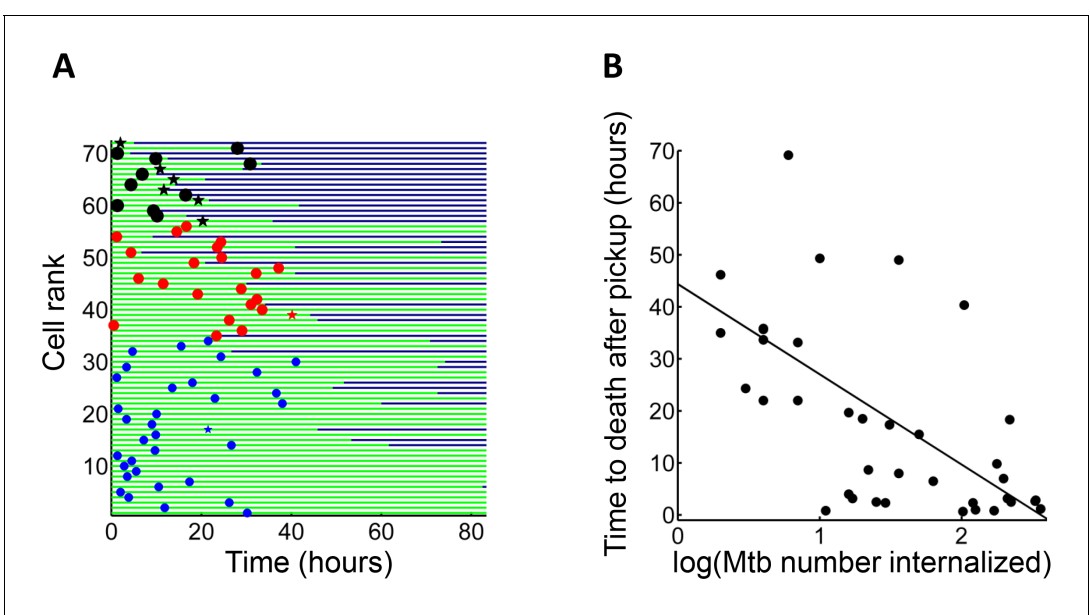

**Figure 3.** Mtb aggregate size determines timing of macrophage death. (**A**) MDMs which internalized exactly one clump before the halfway point of the movie were ranked based on the amount of Mtb internalized (n = 72 cells). The blue circles are internalizations of aggregates of 10 or fewer Mtb, red circles are clumps of 11–49 Mtb, and black circles are >50 Mtb. Line color changes from green to dark blue at time of death. Stars indicate internalizations of dead infected cells. (**B**) Time differences between internalization and death for cells that died as a function of log transformed bacterial number. $R^2$ = 0.56, p=$4\times10^{-8}$(n = 39).

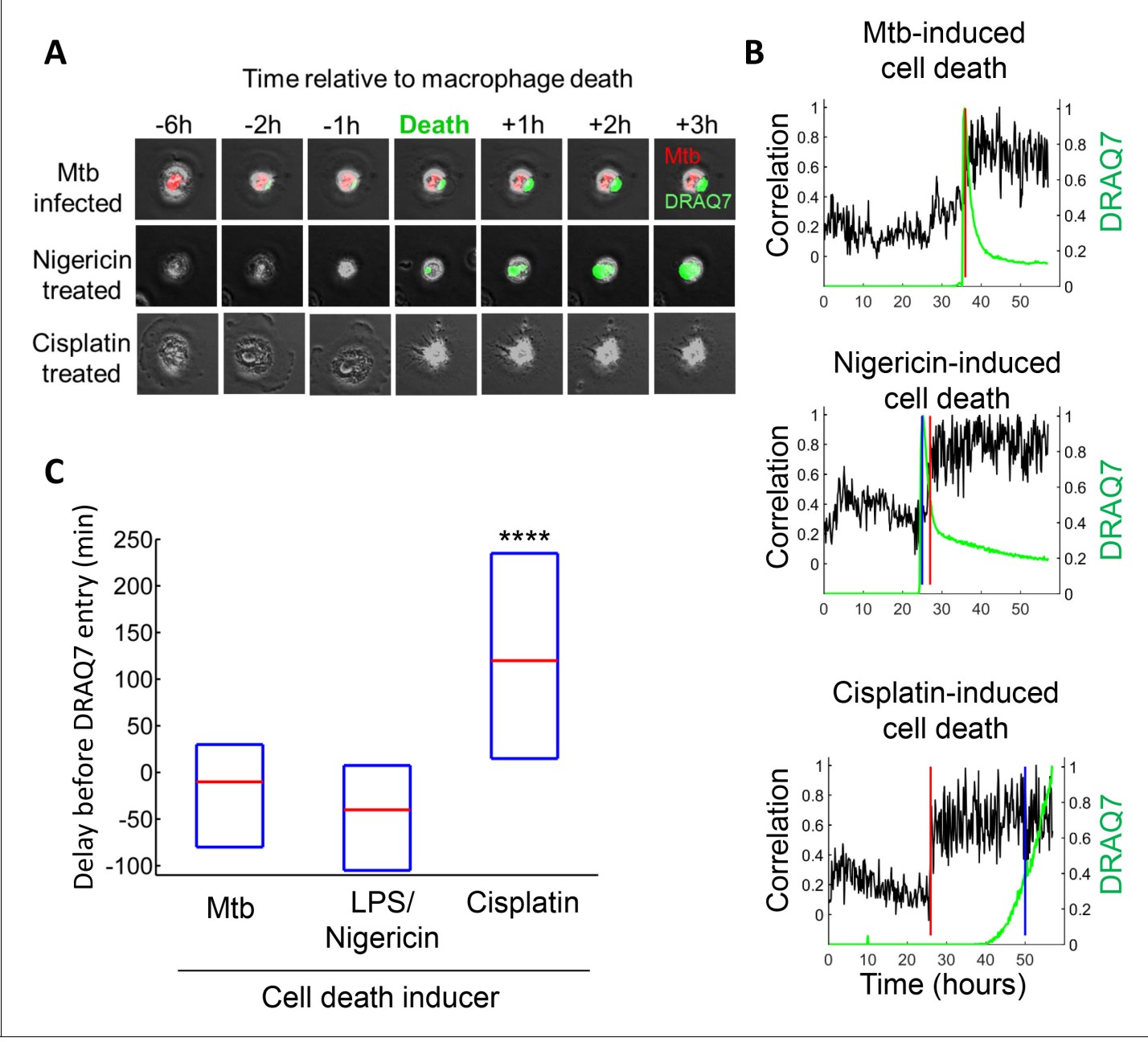

**Figure 4.** Mtb induced macrophage death does not resemble apoptosis. (A) Cell death was caused by internalization of Mtb clumps (top row, Mtb in red), the pyroptosis inducers Nigericin+LPS (middle row), or the apoptosis inducer cisplatin (bottom row). Time of cell death was determined by the cessation of internal movement, and loss of membrane integrity by entry of the fluorescent dye DRAQ7 (green). (B) Quantitation of time of cell death and loss of membrane integrity. Black line is the Pearson correlation of within cell image pixels from one time-lapse frame to the next. Cell death as determined by cessation of internal cell movement is indicated by greater than threshold jump in pixel correlation (red vertical line). Loss of membrane integrity as detected by DRAQ7 is shown by the green line, (normalized to maximum signal) with blue vertical line representing time DRAQ7 levels cross an experimentally determined threshold. In the case of Mtb-induced cell death, the blue and red lines are superimposed. Each graph represents the cell shown in (A). (C) Median time difference (red line, with an interquartile range shown as blue box) between cell death and loss of membrane integrity for cells killed by Mtb (n = 119), 20 μM nigericin and 1 μg/ml LPS (n = 29) or 50 μM cisplatin (n = 76). ****p<0.001 by Kruskal-Wallis rank sum test corrected for multiple hypotheses.

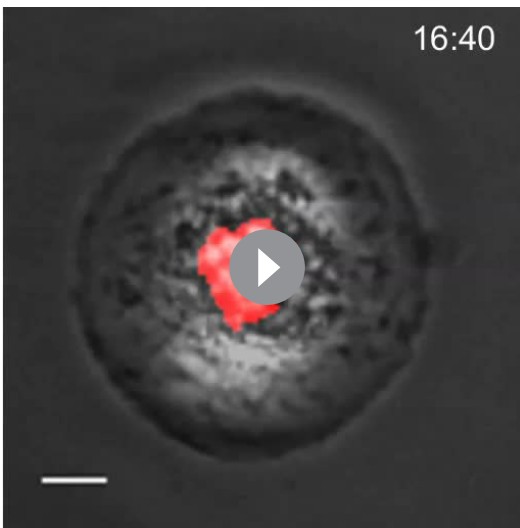

**Video 2.** Death of an Mtb-infected macrophage. MDMs were infected with Mtb H37Rv-RFP (red) and imaged in the presence of the viability dye DRAQ7 (green). The entry of DRAQ7 corresponds to the time of cell death. Scale bar is 20 μm.

had declining Mtb numbers (*Figure 5B*). Intracellular Mtb had a median doubling time, excluding the two negative values, of 24.7 hr (interquartile range 19 to 34 hr, *Figure 5B* inset). The extracellular growth rate of Mtb in the same experiments had a median doubling time of 36.1 hr (interquartile range 31 to 44 hr, *Figure 5B* inset), significantly slower than in dead cells ($p=7\times10^{-8}$, Wilcoxon rank sum test). We confirmed these results by CFU, quantifying the number of Mtb after three days of growth in dead MDM versus extracellular growth in MDM medium. We initiated growth with the same concentrated Mtb culture, which killed approximately 99% of the MDMs after one day, thus allowing Mtb to replicate in the dead macrophages for at least two days (*Figure 5—figure supplement 1A*). We recovered approximately an order of magnitude more Mtb by CFU after growth in dead cells versus after growth in the MDM extracellular medium (*Figure 5—figure supplement 1B*).

We next compared the growth of Mtb in dead cells to that in live infected cells. Live macrophages internalized Mtb throughout the imaging period, and hence an increase in Mtb fluorescence could be the result of either new Mtb internalized, or existing Mtb growing in the live cell. Therefore, we marked the time each macrophage internalized Mtb, and used a 10 hr tracking window which could be fitted between internalization events for many of the cells. We calculated the median fold change in Mtb in this interval (*Figure 6A*). For comparison, we used an interval of the same length in cells after death (*Figure 6B*). In order to investigate the source of variability and the statistical significance of the data, we fitted the temporal dynamics of Mtb fluorescence of each individual infected MDM with an

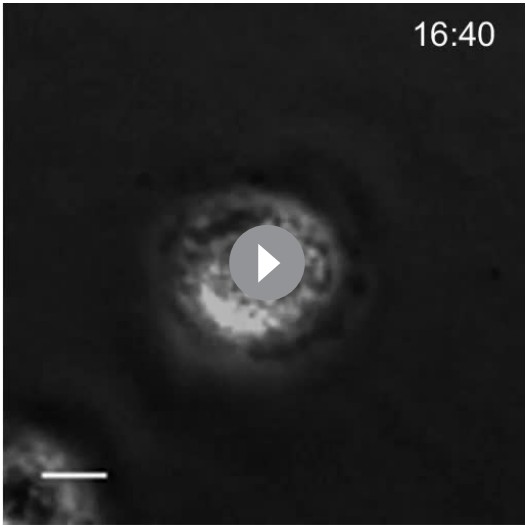

**Video 3.** Pyroptotic death of macrophages. Macrophages were sensitized with 1 μg/ml LPS for 3 hr, then treated with 20 μM nigericin to induce pyroptosis. Cells were imaged in the presence of DRAQ7 (green).

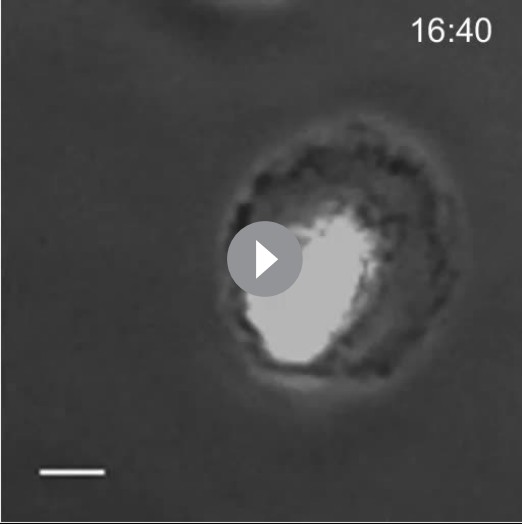

**Video 4.** Cisplatin-Induced cell death. Macrophages were treated with 50 μM cisplatin and imaged in the presence of DRAQ7 (green). Scale bar is 20 μm.

exponential curve. Variability in Mtb growth was evident in individual slopes in live macrophages, with Mtb growing in some cells and decreasing in others (*Figure 6A* inset). Mtb growth in dead cells was less variable (*Figure 6B* inset).

The median growth constant for Mtb in live cells was 0.0012 hour$^{-1}$ (interquartile range −0.016 to 0.022), translating to a doubling time of >100 hr (*Figure 6A*). The slope was not significantly different from 0 (p=0.3, right tailed Sign Test). In contrast, the median growth constant for Mtb in dead cells was 0.033 hour$^{-1}$ (interquartile range 0.021 to 0.045), translating to a doubling time of 21.2 hr (*Figure 6B*). This was faster than growth over the full timescale (*Figure 5B*), possibly indicating faster growth of Mtb in the first hours after host cell death. Mtb growth in dead cells was positive (p=5×10$^{-23}$, right tailed Sign Test), and significantly faster than in live infected cells (p=8×10$^{-15}$, Wilcoxon rank sum test). While the Mtb growth inside live cells may be significantly positive if measured over a longer time period, we can conclude that it is much slower than growth in dead cells.

To understand the mechanism behind the very slow growth in live MDM versus robust growth in dead MDM, we labelled Mtb with the pH detection dye pHrodo prior to MDM infection. We observed that Mtb were in an acidified compartment in live MDM (*Video 6* and *Figure 6C*), with seemingly variable acidification of different bacilli in the same macrophage. Upon Mtb induced MDM death, pHrodo signal was rapidly reduced to near background, indicating that acidification was lost (*Video 6*, *Figure 6D*). This indicates that live infected MDM sequestered Mtb in a phagosomal compartment, which was at least partially acidified. Upon cell death, this compartmentalization disappeared.

We next examined host cell death and Mtb dynamics in several physiological contexts. The immune response to infection activates macrophages to mediate Mtb control (*Russell et al., 2010*; *Means et al., 1999*; *Mosser, 2003*; *Nau et al., 2002*). A key macrophage activator is IFNγ, secreted by several immune cell types (*Chackerian et al., 2001*; *Flesch and Kaufmann, 1987*; *Cooper et al., 1993*; *Flynn et al., 1993*; *Fenton et al., 1997*). We therefore exposed MDM to IFNγ prior to infection and examined their response to internalized aggregates of Mtb. Infection with aggregates led to cell death which, similarly to untreated MDM, resulted in Mtb growth inside the dead infected cells (*Figure 7A*). The median growth constant in dead MDM treated before death with IFNγ was 0.027 hour$^{-1}$ (interquartile range 0.0076 to 0.046), corresponding to a doubling time of 25.8 hr. Alveolar macrophages are responsible for the initial macrophage contact with the pathogen in the lung (*Keane et al., 2000*; *Fenton et al., 1997*; *Nicholson et al., 1996*; *Hirsch et al., 1994*; *Keane et al., 1997*). Similarly to MDM, infection of these cells with aggregates led to cell death followed by Mtb intracellular growth (*Figure 7B*). The median growth constant in dead alveolar macrophages was 0.028 hour$^{-1}$ (interquartile range 0.0014 to 0.055), corresponding to a doubling time of 24.6 hr. We have also quantified the extracellular Mtb growth rate during a ten-hour window (*Figure 7C*). Here, the median growth constant was 0.018 hour$^{-1}$ (interquartile range 0.0015 to 0.034), corresponding to a 39.1 hr doubling time.

Interestingly, Mtb seemed to grow somewhat slower in both IFNγ treated MDM and alveolar macrophages relative to untreated MDM (*Figure 7D*). However, this difference was not statistically significant (p=0.45 for treated versus untreated MDM, and p=0.16 for alveolar versus untreated MDM, Kruskal-Wallis rank sum test corrected for multiple hypotheses). Hence, the growth of Mtb in dead cells seemed to be relatively independent of cellular activation and tissue source.

## Mtb clumps lead to a cascade of macrophage death

We next investigated whether internalization of dead infected cells led to macrophage death. We selected from our dataset all live macrophages, which internalized dead infected macrophages before 90% of the video recording elapsed, to avoid cells where we could not capture the death event because it was too close to the end of the movie. While cell-free pickups of Mtb also occurred in some of these cells, internalization of a dead infected cell was the terminal internalization event in all but two of the observed cells. Frequency of cell death was 89% (*Figure 8*). There was a strong correlation between the time of internalization and death of the internalizing cell (R$^2$ = 0.93, p=8×10$^{-23}$, *Figure 8—figure supplement 1*), with a median time to death of 3.2 hr (interquartile range of 1.5 to 6.0 hr). Chains of macrophage internalizations of dead infected cells, followed by cell death of the internalizing macrophage and uptake by the next cell, were also seen (*Video 7*). We compared the frequency of death from dead cell pickups to the frequency of death in cells

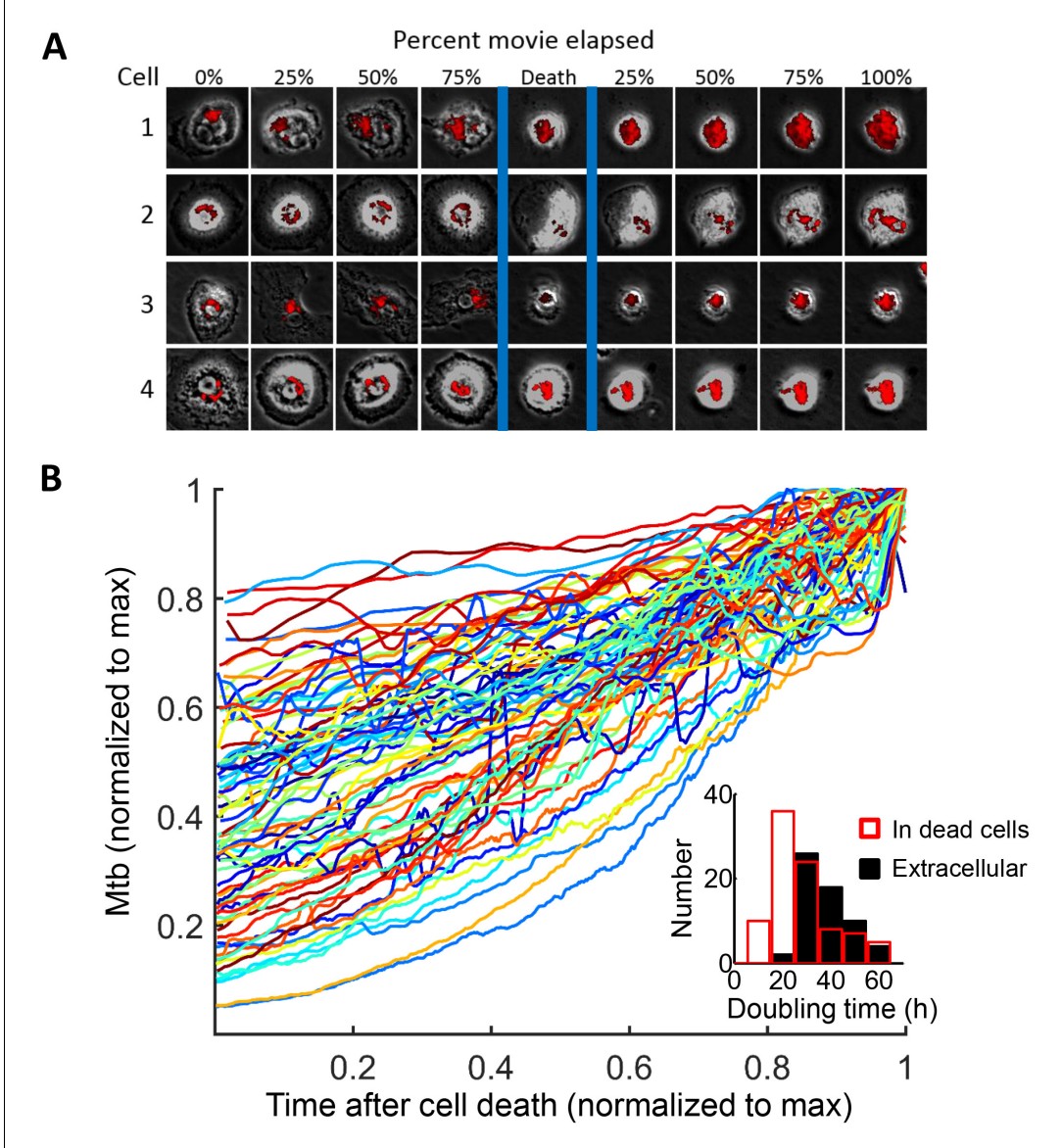

**Figure 5.** Mtb grows robustly inside dead macrophages. (**A**) In silico synchronized images of representative macrophages infected with Mtb and imaged before and after death. Each horizontal set of images represents the same macrophage over time, with the cell death event at the center image. On the left of the cell death event are images of the cell at regular intervals before cell death as a percentage of time the cell was imaged alive. On the right are images at regular intervals after death a percentage of time the cell was imaged dead. (**B**) Traces of Mtb growth in dead cells. Each trace represents the number of Mtb within the same dead MDM and is normalized to its maximum Mtb number and maximum length of time the dead cell was imaged before detachment or movie end. Minimum time for imaging dead cells was 10 hr. In total, 92 dead macrophages were analyzed, with 90 showing positive exponential slopes. The median doubling time for cells with positive slopes was 24.7 hr over five independent experiments. Inset: Doubling times of Mtb in dead cells excluding the two negative values (red, n = 90) and in the extracellular medium (black, n = 60). The median extracellular doubling time was 36.1 hr.

The following figure supplement is available for figure 5:

**Figure supplement 1.** Differences in growth measured by CFU between Mtb in dead cells and extracellular Mtb.

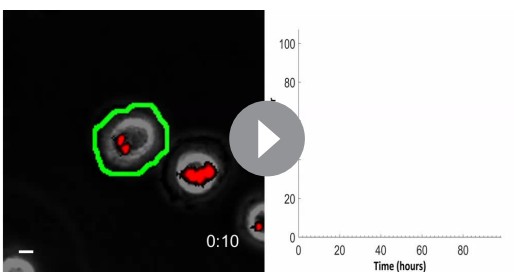

**Video 5.** Mtb grows inside a dead macrophage treated with IFNγ prior to infection. The left panel of the montage is a movie of macrophage infection at a resolution of 10 min between image acquisitions. The border of the analyzed macrophage is shown as a green outline, and the presence of DRAQ7 is shown in blue. RFP-expressing Mtb are in red. Time is hours: minutes. Scale bar is 20 µm. The graph on right shows the fluorescence signal converted to bacterial numbers in the analyzed macrophage. Timing of macrophage death was determined by the lack of internal movement and confirmed by a DRAQ7 entry (vertical line of x's).

internalizing large cell-free aggregates (81–300 bacilli), which had a similar median number of bacteria (dead infected cells contained a median of 114 Mtb, versus 125 Mtb for cell-free aggregates). Death frequency was 81% for cells that internalized cell-free aggregates, and the difference between death frequencies was not significant (p=0.16). This indicated that the highly cytotoxic effect of dead infected cell internalization can be mostly explained by the high number of Mtb they contain.

If Mtb aggregates kill the phagocyte, remain associated with it, and grow in the dead cell, each new macrophage joining the infection chain should be faced with a greater or equal probability of cell death due to increasing Mtb numbers (*Figure 9A*, *Video 7*). To model the consequences of host cell death associated with bacterial growth, we performed numerical simulations to determine the number of dead host cells through time. Here, we do not claim to capture the exact dynamics of what occurs in vivo in the diseased lung, but rather illustrate the trend (see Materials and methods for simulation details). We compared the number of dead host cells when the Mtb doubling time in dead cells was one day, versus when it was two days (where two days is within the range of the doubling times observed for extracellular Mtb, which was 22–75 hr for the data presented in the inset of *Figure 5B*). As in the experimental results, host cell death was dependent on the bacterial number internalized, and host cells were restrictive of intracellular bacterial growth when alive and permissive when dead. A one day doubling time yielded substantially higher numbers of dead cells in many of the simulations (*Figure 9B*, left panels). Strikingly, only the one day doubling time in dead cells resulted in high Mtb growth over the simulation period of 60 days (*Figure 9B*, right panels). High Mtb growth was very sensitive to the doubling time (*Figure 9—figure supplement 1*), and this could not be explained by the differences in doubling time if growth was cell-free (*Figure 9B*, white lines in right panels), as both growth rates were rapid for the timescale of the simulations. Rather, this simplified simulation shows the presence of positive feedback in the host cell-Mtb interactions: faster Mtb growth in the dead cell leads to larger intracellular clumps upon phagocytosis by another cell, and hence to a higher probability of death of the phagocytosing cell, leading to Mtb growth in the new dead cell and a higher probability to kill the next phagocyte in the infection chain.

## Discussion

Active Mtb infection and pulmonary disease involve host cell necrosis as well as extensive bacterial proliferation, both necessary for infectious aerosol formation and transmission to the next host (*Russell, 2007*; *Russell et al., 2010*). While in general Mtb bacilli are hard to find in intact closed granulomas (*Lin et al., 2014*), high Mtb concentrations have been reported in vivo upon examination of the cavity surface in human tuberculous granulomas (*Hunter, 2011*; *Hunter et al., 2007*; *Welsh et al., 2011*) as well as cavity containing granulomas in rabbits infected with experimental pulmonary TB (*Kaplan et al., 2003*), and the cavity surface of granulomas with a necrotic core in the granuloma-forming Kramnik mouse model (*Irwin et al., 2015*). Histologic examination has shown clumped bacteria as well as highly infected macrophages at the luminal surface of cavities in areas with extensive cellular necrosis (*Kaplan et al., 2003*; *Hunter, 2011*; *Irwin et al., 2015*). While it can be inferred that high bacillary loads cause host cell death, this view is incomplete, as it lacks insight into the dynamic interactions between host cells and bacilli which cause the buildup of bacterial numbers and host necrosis at a particular site. Furthermore, since Mtb clumps are present in vivo, it

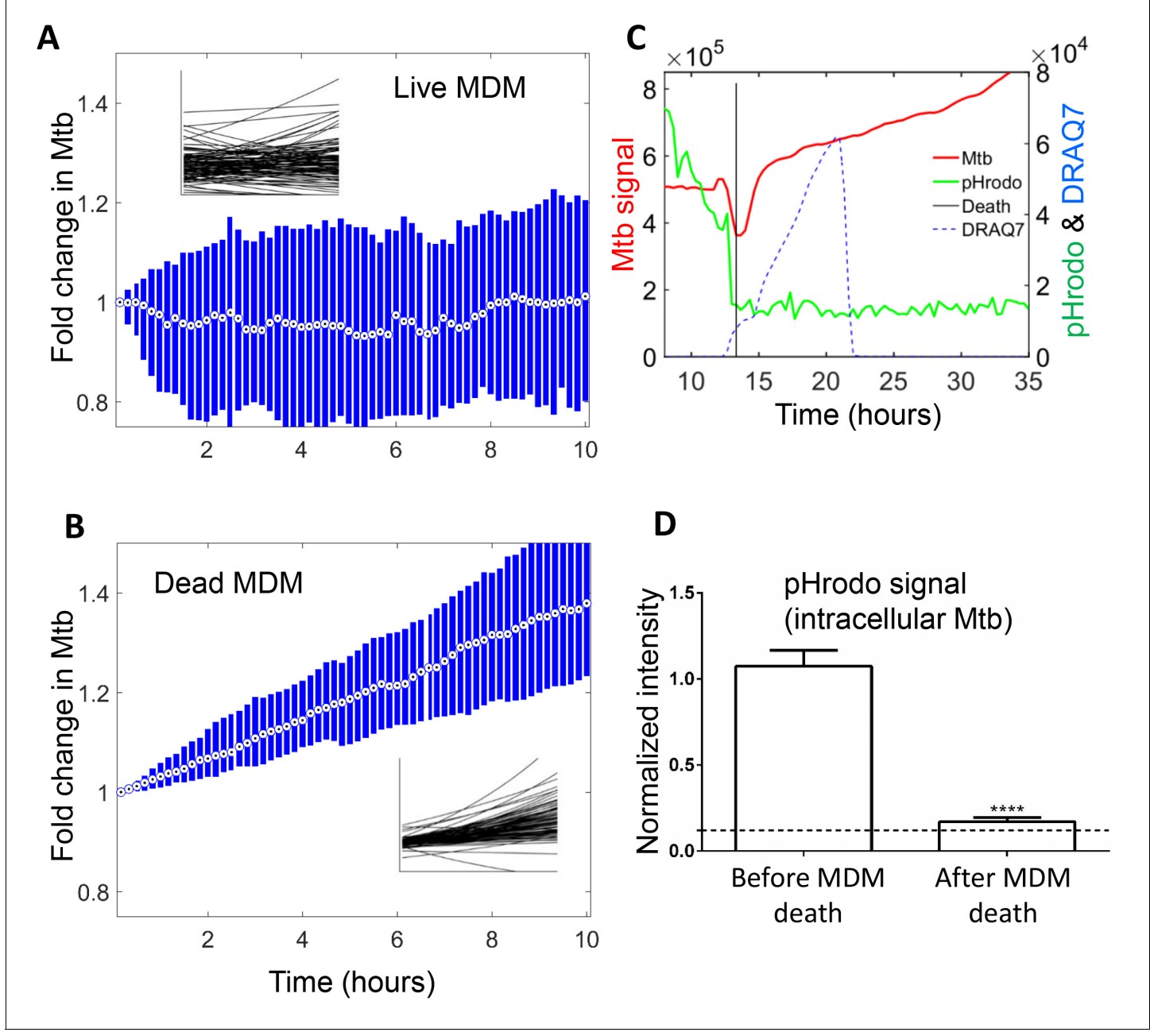

**Figure 6.** Mtb growth in macrophages before and after macrophage death. Median (circle with dot) and interquartile range (blue rectangle) of Mtb signal in live (A) and dead cells (B). Insets show exponential fits of Mtb dynamics in the individual live (n = 101) or dead (n = 92) macrophages. X and y-scales on insets same as on main panels. The median doubling time of Mtb was >100 hr in live cells, 21.2 hr in dead cells. (C) MDMs were infected with Mtb-mCherry labelled with pHrodo Green, which fluoresces in low pH. Mtb (red line), pHrodo (green line) and DRAQ7 (dashed blue line) signals of a representative MDM tracked by time-lapse microscopy is shown. The time of cell death, as determined by DRAQ7 entry, is indicated as a vertical black line (raw data provided in *Figure 6—source data 1*). (D) pHrodo signal was quantified 3 hr before and following death of each cell. Means and standard error of cells (n = 74 cells) combined from two independent experiments. The dashed horizontal line represents signal threshold. ****p<0.0001, paired t-test of pHrodo signal before and after cell death.

The following source data is available for figure 6:

**Source data 1.** pHrodo signal over time.

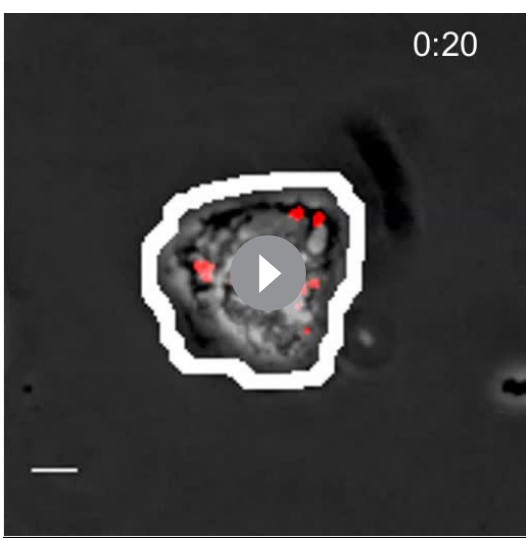

**Video 6.** Loss of phagosome acidification upon Mtb induced host cell death. Mtb were stained with the pH detection dye pHrodo at 100 µM and used to infect MDM in the presence of the cell death indicator dye DRAQ7. pHrodo fluorescence is shown in green, mCherry-expressing Mtb in red, and DRAQ7 fluorescence in blue. Time is hours:minutes. Scale bar is 20 µm.

is necessary to consider the effect of clumps on the dynamics of the host-pathogen interaction (*Orme, 2014*; *Sani et al., 2010*). This entails using non-detergent treated Mtb capable of aggregate formation. Here we used time-lapse microscopy of detergent untreated Mtb infection of primary human macrophages to map out the interactions between Mtb and the host macrophages. We quantified host cell-Mtb dynamics over 80 hr of growth at a 10 min resolution between time-lapse frames, necessary to capture several replication cycles of this slow growing pathogen. To our knowledge, this study is the first to use long duration time-lapse microscopy combined with automated image analysis to decipher Mtb infection dynamics of primary human macrophages.

We confirmed at the single cell level that the probability of macrophage death increased with multiplicity of infection (*Figure 1*, *Video 1*), consistent with a previous report of a threshold of pathogens which a macrophage can control in *Trypanosoma cruzi* infection (*Tanaka et al., 1982*) as well as the previously found increase of cell death with increasing Mtb load (*Lee et al., 2006*; *Repasy et al., 2013*). The death pathway was non-apoptotic (*Figure 4*), consistent with previous reports of host cell death at high multiplicities of Mtb infection (*Lee et al., 2006*; *Repasy et al., 2013*). However, in addition to the multiplicity of infection, host cells were found to be sensitive to the aggregation state of the bacilli: pickup of a single large aggregate was more cytotoxic than pickup of several smaller aggregates with a similar total number of Mtb (*Figure 2*). After internalization of large clumps, macrophage death was rapid (*Figure 3*, *Videos 1*, *5* and *7*). The observed rapid breakdown of host cell membranes with Mtb induced cell death (*Videos 2* and *5*, *Figure 4*) is consistent with reports of Mtb being present in the cytoplasm, as opposed to the phagosome (*Russell, 2001*), of non-apoptotic cells (*van der Wel et al., 2007*; *McDonough et al., 1993*; *Simeone et al., 2012*). If breach of the phagolysosomal membrane precedes death in Mtb infection, one reason for the increased cytotoxicity of aggregates may include increased membrane breaching (*Rodríguez-Muela et al., 2015*; *Sargeant et al., 2014*; *Boya and Kroemer, 2008*). In addition, Mtb induced mitochondrial membrane breakdown (*Duan et al., 2002*), interference with host plasma membrane repair (*Divangahi et al., 2009*, *2010*), and toxin secretion (*Sun et al., 2015*) would all be expected to scale with Mtb number per macrophage.

Host cell death did not eliminate intracellular Mtb (*Figure 5*), consistent with previous reports at the cell population level showing that Mtb induced necrosis does not clear the bacilli (*Fratazzi et al., 1999*; *Molloy et al., 1994*; *Lee et al., 2006*; *Park et al., 2006*; *Dobos et al., 2000*; *Duan et al., 2002*; *Divangahi et al., 2009*, *2010*; *Sun et al., 2015*; *Tobin et al., 2012*), and that upregulation of *Ipr1*, which switches the host response to Mtb from necrosis to apoptosis in mice, limits Mtb growth (*Pan et al., 2005*). Pre-stimulation with IFNγ or use of alveolar macrophages instead of MDM did not substantially alter the ability of Mtb to survive host cell death. These observations differ from results showing that efferocytosis leads to the elimination of Mtb (*Martin et al., 2012*). However, the protective effect of efferocytosis may occur when cells die by apoptosis and the number of bacilli inside the dead cell is small while the number of uninfected macrophages is large, enabling the surrounding macrophages to divide up the bacillary load of a single dead infected cell. This was not the case in our experiments.

The observation that Mtb growth was slowest in live cells, intermediate in the extracellular environment, and most robust in dead infected cells (*Figures 5–7*), suggest that the dead cell may

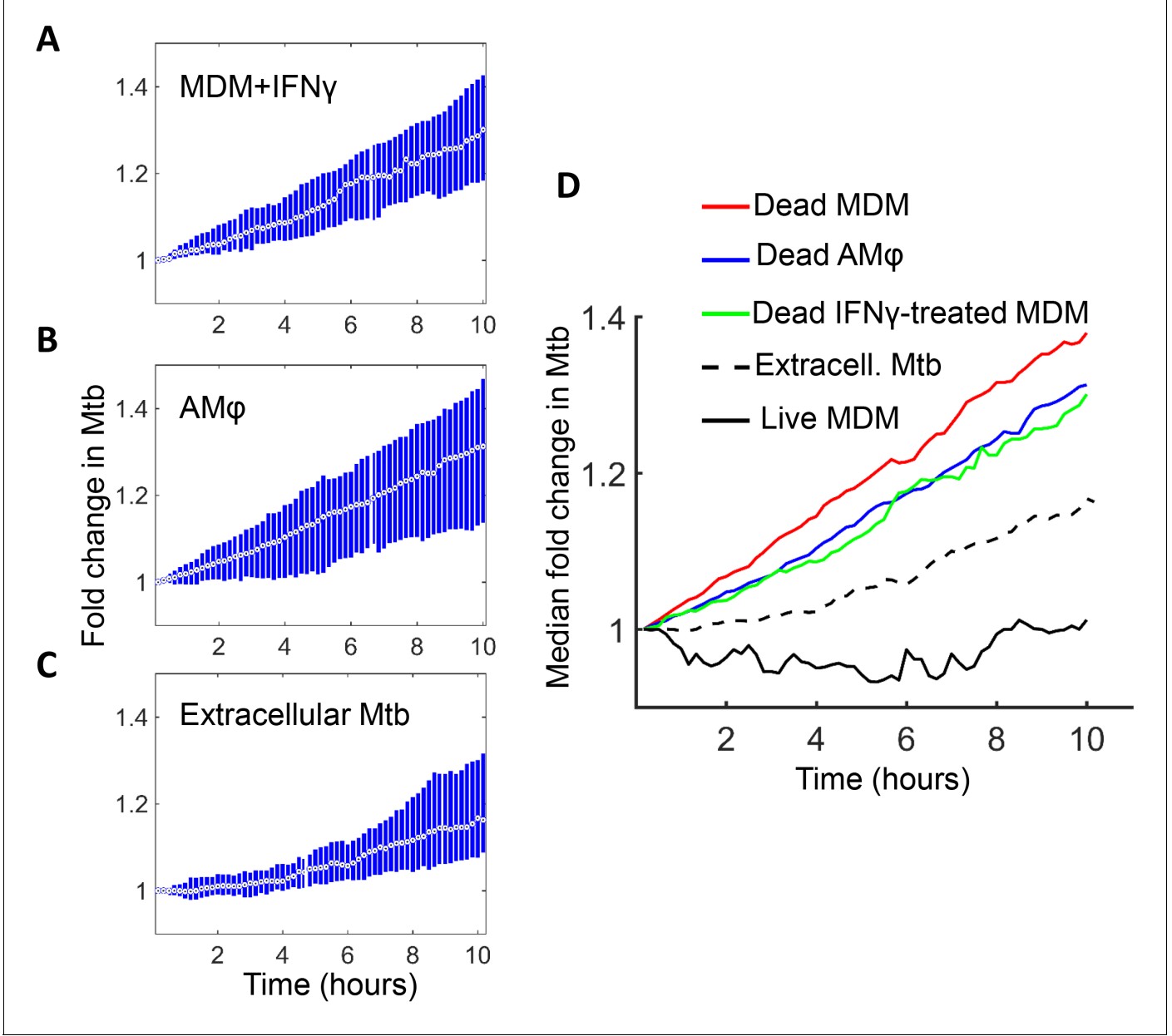

**Figure 7.** Mtb grows in IFNγ treated MDM and alveolar macrophages after cell death. (**A**) MDM was exposed to IFNγ for 18 hr, then infected with Mtb and imaged (raw data provided in *Figure 7—source data 1*). Shown is the median fold change (circle with dot) and interquartile range (blue rectangle) of Mtb in dead infected cells. (**B**) Median fold change and interquartile range of Mtb in dead infected human alveolar macrophages (AMφ) isolated from bronchoalveolar lavage and infected in vitro (raw data provided in *Figure 7—source data 2*). (**C**) Median fold change in Mtb number in extracellular aggregates of Mtb over a 10 hr timespan (raw data provided in *Figure 7—source data 3*). (**D**) Combined plot of median values for all Mtb growth conditions. The number of dead cells or extracellular clumps analyzed: n = 49 (IFNγ treated MDM), n = 220 (alveolar macrophages), n = 60 (extracellular clumps). Non-IFNγ treated live and dead MDM numbers are as in *Figure 6*.

The following source data is available for figure 7:

**Source data 1.** Intracellular Mtb fluorescence through time in dead IFNγ treated MDM.
**Source data 2.** Intracellular Mtb fluorescence through time in dead alveolar macrophages.
**Source data 3.** Extracellular Mtb fluorescence through time.

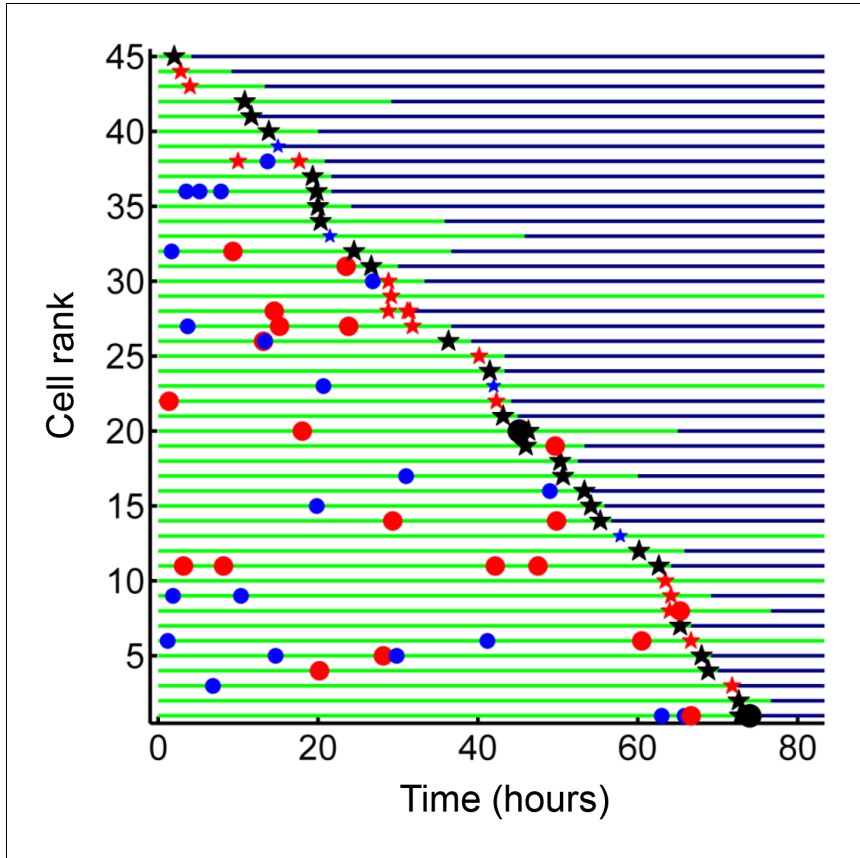

**Figure 8.** Internalization of dead infected cells leads to rapid cell death. Cells are ranked according to the time of last pickup, with cells ranked highest internalizing earliest. The frequency of cell death was 89%. The blue circles are internalizations of aggregates of 10 or fewer Mtb, red circles are clumps of 11–49 Mtb, and black circles are >50 Mtb. Line color changes from green to dark blue at time of death. Stars are internalizations of dead infected cells.

The following figure supplement is available for figure 8:

**Figure supplement 1.** Correlation of time of last pickup and time to death in the cells internalizing a dead infected cell.

provide Mtb with a favorable growth niche relative to the intact phagosome in live cells or the extracellular environment, while live cells provide a poor growth environment either because of antimicrobial effector mechanisms or reduced access to nutrients in the Mtb compartment. Our results indicating acidification of Mtb (*Figure 6C–D*) are consistent with the presence of antimicrobial effector mechanisms that reduce Mtb growth in Mtb compartmentalized to the phagosome. Mtb can inhibit phagosome-lysosome fusion and complete phagosome acidification (*Vergne et al., 2003*; *Rohde et al., 2007*; *Via et al., 1997*). However, acidification may occur when macrophages are activated with cytokines such GM-CSF (*Denis and Ghadirian, 1990*), as are the MDMs used for this study. In this case, Mtb growth may be impaired or arrested in acidified vacuoles (*Gomes et al., 1999*). Acidification was lost immediately upon cell death (*Video 6*, *Figure 6C–D*). However, loss of antimicrobial effector mechanisms may not be the only factor responsible for robust growth in dead cells, as growth is faster than in the extracellular medium, indicating that Mtb may have access to cytosolic nutrient sources, which promote growth in other intracellular pathogens (*Appelberg, 2006*; *Ray et al., 2009*). If Mtb growth in the dead cells was solely dependent of leakage of nutrients from the extracellular environment, the intracellular bacilli would not be expected to replicate any faster than extracellular Mtb. This is clearly not what we observe.

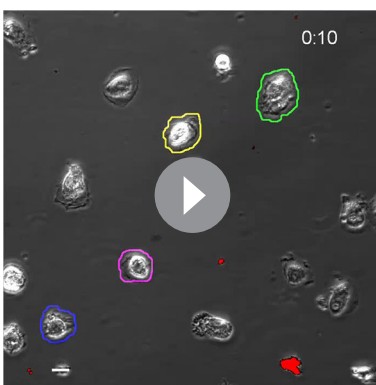

**Video 7.** Cell death cascade following phagocytosis of a dead infected macrophage. The borders of four macrophages are shown as green, yellow, magenta, or blue outlines. RFP-expressing Mtb are shown in red. Time is hours:minutes. Scale bar is 20 µm.

In these experiments we used a pure macrophage culture to investigate the outcome of macrophage-Mtb interactions. In the granuloma, the presence of other cell types can influence outcomes. Particularly relevant are the T cell subsets of the adaptive cellular immune response (*Cooper and Flynn, 1995*; *Flynn et al., 1995*, *1992*; *Mogues et al., 2001*). T cells control Mtb by activating macrophages to kill Mtb through IFNγ secretion (*Flynn et al., 1993*), and CD8[+] cytotoxic T cells target intracellular bacteria by granulysin release (*Stenger et al., 1998*), which has been reported to enable granzymes to enter and kill the bacteria (*Walch et al., 2014*). These mechanisms would counteract the spread of Mtb, but would likely be restricted by the Mtb mediated death of the host cell before recognition and targeting by T cells, limited T cell effector function in the granuloma (*Egen et al., 2011*), lack of T cells in the inner cell layer of granulomas (*Ulrichs et al., 2004*), and impaired expression of perforin and granulysin observed in Mtb granulomas (*Andersson et al., 2007*).

The process we observe likely reflects a positive feedback loop for Mtb replication in late stage infection, as illustrated by the outcomes of the simplified model in *Figure 9*. At this infection stage, localized expansion of Mtb numbers may occur by the following scenario, captured in *Figure 8* and strikingly in *Video 7*: serial killing is initiated when a human macrophage internalizes a clump of bacilli which causes its death by necrosis. Upon death, Mtb rapidly grows in the dead cell, with a doubling time faster than either in the extracellular environment or in live cells (where growth is minimal). The next macrophage to internalize the Mtb, this time encased in a dead cell, faces a larger Mtb clump and also dies. The new dead infected cell now provides fuel for the next round of bacterial growth and bait for the next macrophage. This illustrates how, once initiated, Mtb replication can be locally stabilized in the active state.

## Materials and methods

### Ethical statement

Blood was obtained from adult healthy volunteers after written informed consent (University of KwaZulu-Natal Institutional Review Board approval BE022/13). Alveolar macrophages were obtained from bronchoalveolar lavage as part of an indicated diagnostic procedure after written informed consent (University of KwaZulu-Natal Institutional Review Board approval BE037/12).

### Macrophage cultures

Buffy coats from HIV-negative blood bank donations were obtained from the South African National Blood Service (Human Research Ethics Committee approved protocol IRB00007553) or a cohort of healthy donors (UKZN Institutional Review Board approval BE022/13). Informed consent was obtained. Peripheral blood mononuclear cells were isolated by density gradient centrifugation using Histopaque 1077 (Sigma-Aldrich, St Louis, MO). CD14[+] monocytes were purified under positive selection using anti-CD14 microbeads (Miltenyi Biotec, San Diego, CA). 2 ml of $10^5$/ml monocytes were added to 0.01% fibronectin (Sigma-Aldrich) coated 35 mm glass bottom optical dishes (Mattek, Ashland, MA) and differentiated in macrophage growth medium containing 1% each of HEPES, sodium pyruvate, L-glutamine, and non-essential amino acids, 10% human AB serum (Sigma-Aldrich), and 50 ng/ml GM-CSF (Peprotech, Rocky Hill, NJ) in RPMI. The cell culture medium was changed one day post plating and half the media was replaced on day 3 and 6 post plating. To isolate alveolar macrophages, bronchoalveolar lavage fluid was centrifuged at 300× g for 10 min, and the cell pellet washed twice with phosphate buffered saline (PBS). The cells were counted, and adjusted to

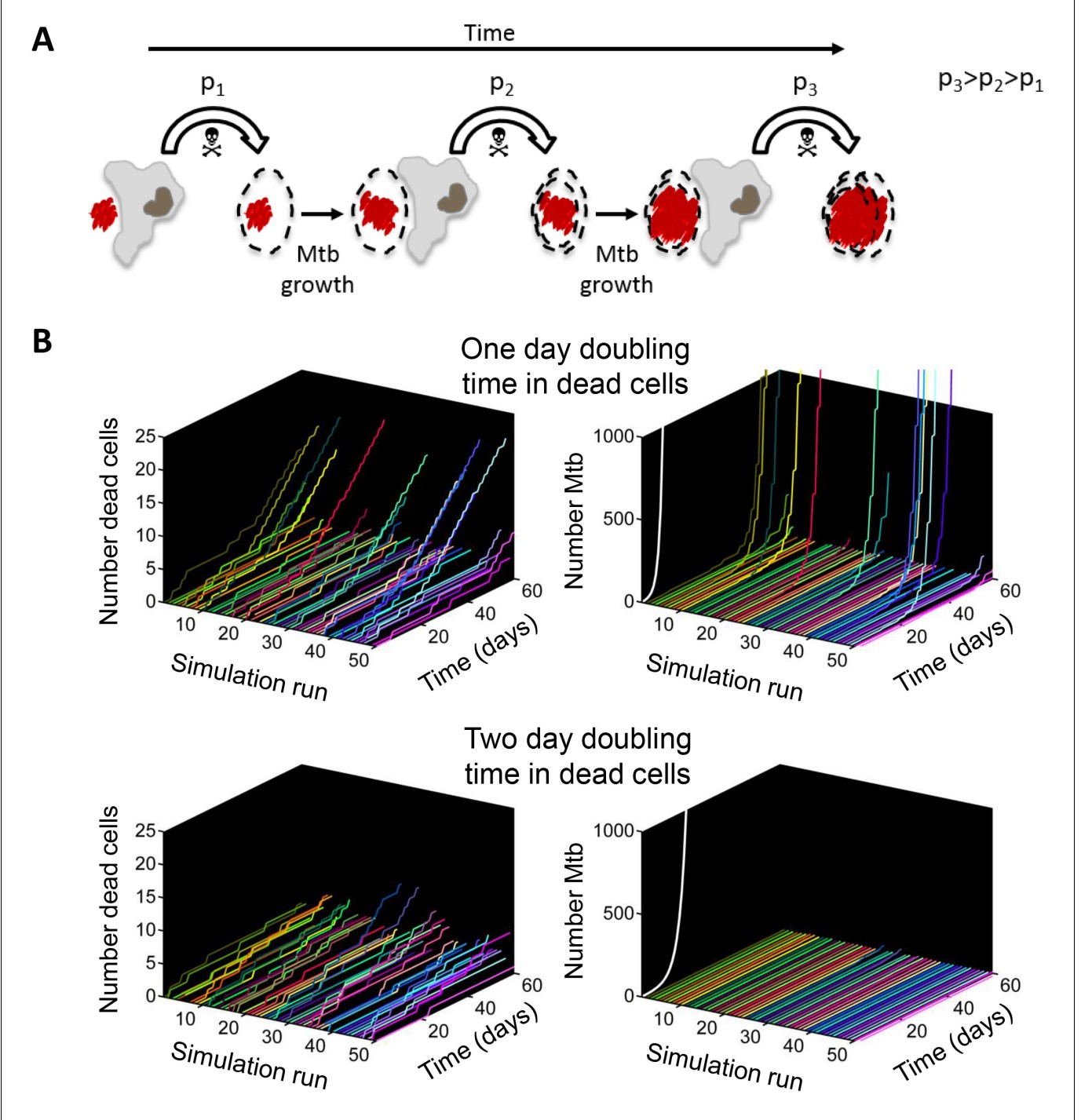

**Figure 9.** Positive feedback in Mtb infection. (A) Schematic of the positive feedback loop. Mtb are in red, live cells are in grey and dead cells are outlined by dashed borders. Time is from left to right. Probabilities of death at time points 1, 2, and three for macrophages internalizing dead infected cells are represented by $p_1$, $p_2$, and $p_3$, where $p_3 > p_2 > p_1$ due to Mtb growth in the dead cells. (B) Results of numerical simulations of the number of dead host cells over time and the corresponding numbers of accumulated Mtb given a one day (top panels) or two day (bottom panels) doubling time inside dead infected host cells. Each line represents an independent simulation (numbered on the x-axis), time in days is on the z-axis, and the cumulative number of dead cells or number of Mtb is on the y-axis. Each run was initiated with the internalization of one bacillus inside a dead cell. White line represents Mtb extracellular growth given a one or two day doubling time. The fraction of simulations reaching 1000 or more Mtb within 60 days was 0.21 for a one day doubling time and 0.0003 for a two day doubling time (code provided as *Source code 3*).

*Figure 9 continued on next page*

*Figure 9 continued*

The following figure supplement is available for figure 9:

**Figure supplement 1.** Sensitivity of Mtb expansion to doubling time in dead cells.

$10^5$ cells/ml using macrophage growth media with 2.5 µg/mL amphotericin B to inhibit potential fungal contamination. The cells were plated in glass bottom dishes overnight, followed by three washes with PBS, and the media was replaced.

## Mtb cultures

The RFP fluorescent strain of H37Rv Mtb was derived by transforming the parental strain with a plasmid with the TagRFP gene downstream of the constitutive groEL promoter (gift from D. Russell). The mCherry fluorescent strain of H37Rv Mtb was derived by transforming the parental strain with a plasmid with mCherry under the smyc' promoter (gift from D. Russell). Mtb were maintained in Difco Middlebrook 7H9 medium enriched with oleic acid-albumin-dextrose catalase supplement (BD, Sparks, MD). Three days before macrophage infection, Mtb were switched to grow in Tween 80-free media. On the day of infection, exponentially growing bacterial culture was pelleted at 2000 × g for 10 min, washed twice with 10 ml PBS, and large aggregates broken up by shaking with sterilized 2–4 mm glass beads for 30 s. 10 ml of PBS was added and large clumps were further excluded by allowing them to settle for 5 min. An inoculum of 1 ml was taken from the center of the suspension for macrophage infections.

## Mtb infection

MDM and alveolar macrophages were inoculated with 5 µl Mtb suspension (MOI ~5 by CFU) and either imaged after 2 hr to allow the Mtb to settle, or incubated 18 hr, wash five times to remove extracellular bacteria and then imaged.

## Time-lapse microscopy

Macrophages and bacteria were imaged using an Andor integrated (Andor, Belfast, UK) Metamorph-controlled (Molecular Devices, Sunnyvale, CA) Nikon TiE motorized microscope (Nikon Corporation, Tokyo, Japan) with a 20x, 0.75 NA phase objective. For Mtb RFP and mCherry fluorescence, excitation source was a 561 laser line and emission was detected through a Semrock Brightline 607 nm filter (Semrock, Rochester, NY). Images were captured using an 888 EMCCD camera ((Andor). Temperature (37°C), humidity and $CO_2$ (5%) were controlled using an environmental chamber (OKO Labs, Naples, Italy). Approximately 40 fields of view were captured every 10 min, one phase contrast image and one fluorescent image per field at every time point. Fluorescence readings after death were confirmed by widefield microscopy (data not shown), but fluorescence readings before and particularly at the point of cell death were strongly influenced by cell movement in the z-plane and are not included in the analysis. For imaging data after cell death, the fluorescence signal starting 4 hr after the cell death event was used for analysis. For DRAQ7 (BioStatus, Leicestershire, UK) entry, excitation source was a 640 nm laser and emission collected through a Semrock Brightline 685 nm filter For pH dye pHrodo detection, excitation source was a 488 laser while emission was collected through a Semrock 525 nm filter.

## Determination of cell borders and conversion of Mtb fluorescence to bacterial number

To determine cell borders, phase contrast images were segmented by a custom code (*Source code 1*) using the Matlab R2014a image analysis toolbox. Cells were identified using edge detection and cell borders between adjacent cells discriminated using the watershed algorithm. Segmentation was manually curated for errors. Cells in each frame in the same field of view were linked to cells in the previous frame by root mean squared closest centroid. To determine the number of Mtb internalized per cell, single Mtb fluorescence was obtained by filtering the culture through a 5 µm filter and validating the resulting product as single cell by microscopy. We converted Mtb fluorescent signal inside cells to bacterial numbers by dividing the signal by mean single Mtb fluorescence at the

acquisition settings used for each experiment. Each macrophage internalization event was manually segmented at the time of pickup, and the fluorescence from the internalized bacteria was divided by the movie specific average fluorescence per bacterium to obtain Mtb number per internalization.

## Validation of Mtb fluorescence by CFU quantitation

To confirm that fluorescence measurements by microscopy reflected actual Mtb expansion, we grew fluorescently labeled-Mtb as a suspension in 7H9 media and sampled aliquots of the suspension by imaging or plating on 7H10 agar plates over a three day period.

## Induction of cell death

Macrophages were either infected with H37Rv Mtb, treated with 50 µM cisplatin (Sigma-Aldrich), or primed with 1 µg/ml *Escherichia coli* LPS (Sigma-Aldrich) for 3 hr, washed, and then incubated with nigericin (20 µM, Sigma-Aldrich). Cells were then incubated with 3 µM DRAQ7 and imaged as described above.

## MDM activation with IFNγ

After seven days of differentiation, MDM was treated with 200 U/ml IFNγ (Peprotech) or vehicle control (0.01% BSA) in fresh culture media, incubated for 18 hr, then infected with Mtb and imaged in the presence of IFNγ as described above for non-IFNγ treated MDM. Activation of macrophages by IFNγ was confirmed by flow cytometry for the upregulation of HLA-DR and CD86 using anti-HLA-DR conjugated to Alexa Fluor 488 and anti-CD86 conjugated to Alexa Fluor 647 monoclonal antibodies (Biolegend, San Diego, CA).

## Determination of cell death

Time of cell death was determined by an image analysis algorithm detecting cell detachment or cessation of internal movement. For the cessation of internal movement, a 20 by 20 pixel square around the centroid of the cell at the last frame was used. The pixels in the square in one frame were compared to the pixels in the same square on the previous frame. Given row r and column c coordinates, for all pixels $X_{r,c}$ in frame i and all corresponding pixels $Y_{r,c}$ in frame i−1, the Pearson's Correlation Coefficient was computed as $\frac{\sum (X_{r,c} - \bar{X})(Y_{r,c} - \bar{Y})}{\sqrt{\sum (X_{r,c} - \bar{X})^2}\sqrt{\sum (Y_{r,c} - \bar{Y})^2}}$. The correlation between frames in cell interiors in dead cells was high, while internal organelle movement in live cells kept correlation low between frames. The time of transition between a living and dead cell was determined by fitting two horizontal lines to the data. If the mean of the line corresponding to the later time points was greater than 1.5-fold the mean of the line for the earlier time-points, the time point at which the transition from the first to second line occurred was the time of death. The fold change threshold was selected to maximize sensitivity and specificity relative to death determined by DRAQ7.

## CFU determination of Mtb growth in dead macrophages

MDMs were infected with Mtb at MOI ~30, six-fold higher than the MOI used in other infections and sufficient to cause >99% macrophage death after one day. After three days, MDMs were lysed with 0.1% Triton X, and the suspensions plated in duplicate on Difco Middlebrook 7H10 agar enriched with oleic acid-albumin-dextrose catalase supplement (BD) to quantify Mtb growth in comparison to Mtb grown in parallel in macrophage media without MDMs.

## pHrodo labeling

Mtb was labelled with 100 µM pHrodo Green STP Ester (Thermo Fisher, Waltham, MA) in 100 mM sodium bicarbonate buffer, pH 8.5 for 30 min at room temperature, and then washed three times with phosphate buffered saline before proceeding to infection of MDMs. The DNA intercalating dye DRAQ7 was used at 3 µM in the same experiments to detect cell death.

## Statistical analysis

Where noted, bootstrap (*Efron and Tibshirani, 1994*) was used to determine statistical significance. p-values of differences between death frequencies between two samples x and y, with Freq(death)$_x$ > Freq(death)$_y$, were determined using Matlab 2014a by randomly selecting with replacement the

same number of cells from x, one at a time, as contained in y. The frequency of death in this randomly selected set of cells was determined, and the procedure repeated 10,000 or 100,000 times to create vector $F_{rand}$. The p-value was calculated as the number of elements in $F_{rand}$ smaller than Freq $(death)_y$, divided by the number of times the procedure was repeated. In the case of multiple comparisons, the significance threshold was made more stringent by adjusting for the number of comparisons by the Bonferroni method (**Noble, 2009**). Statistical analysis not using bootstrap was performed using Graphpad Prism 6 and identified in the figure legends.

### Simulation of Mtb dynamics

The simulation determined the total number of dead host cells per unit time $N_D(t)$ and number of Mtb per unit time $m(t)$ as a function of Mtb doubling time ($t_d$), where time $t$ is in increments of one day. A doubling time of one day was what was experimentally measured in dead cells. To compare to a slower doubling time, for simplicity we used a value of two days, which was within the physiological range of Mtb growth (for example, the doubling time range for extracellular Mtb is 22–75 hr for the data shown in the inset of *Figure 5B*). Each run was initiated with the internalization of 1 Mtb. Two approximations were made based on our experimental results: (1) No Mtb growth occurred inside live cells. (2) Probability of death after internalization of a dead infected cell depended only on the number of Mtb internalized. The probability of cell death $P_D(m)$ per day after Mtb internalization was determined experimentally from the time-lapse data and binned for simplicity for Mtb numbers $m \leq 10$, $10 < m \leq 100$, and $m > 100$. Death probabilities were determined from the data to be $P_D(m \leq 10)=0.06$, $P_D(10 < m \leq 100)=0.50$, and $P_D(m > 100)=0.93$. To implement the cell death decision, a random number from a uniform distribution was drawn at each time increment. If the random number was less than or equal to $P_D(m)$, then $N_D(t + 1)=N_D(t)+1$, else $N_D(t + 1)=N_D(t)$. After cell death at time $t$, there was a one day delay before the next pickup, and $m(t + 1)= m(t)e^r$, where $r = ln2/t_d$. Simulation script (Matlab) is attached as supplementary material. To examine the sensitivity of the number of Mtb after 60 days to doubling time, we ranged $t_d$ from 1 to 3 days, and calculated the fraction of the simulations in which the number of Mtb expanded by more than three orders of magnitude (from 1 to >1000 bacilli).

## Acknowledgements

This study was supported by a Bill and Melinda Gates Foundation Award OPP1116944. AS was supported by the Human Frontiers Science Program Career Development Award. We thank Dr. Emily Wong for facilitating and maintaining the bronchoalveolar lavage cohort.

## Additional information

### Funding

| Funder | Grant reference number | Author |
|---|---|---|
| Sub-Saharan African Network for TB/HIV Research Excellence | | Kelly Pillay |
| Burroughs Wellcome Fund | Burroughs-Wellcome Fund/ American Society of Tropical Medicine and Hygiene fellowship | Emily B Wong |
| American Society of Tropical Medicine and Hygiene | Burroughs-Wellcome Fund/ American Society of Tropical Medicine and Hygiene fellowship | Emily B Wong |
| National Institutes of Health | K08 AI118538 | Emily B Wong |
| Bill and Melinda Gates Foundation | OPP1116944 | Alex Sigal |
| Human Frontier Science Program | CDA00050/2013-C | Alex Sigal |

The funders had no role in study design, data collection and interpretation, or the decision to submit the work for publication.

## Author contributions

DM, Conceptualization, Data curation, Software, Formal analysis, Validation, Investigation, Visualization, Methodology, Writing—original draft, Project administration, Writing—review and editing; MB, YG, CM, Investigation, Methodology; CMA, Data curation, Investigation, Methodology; SS, LO, Software, Methodology; OC, KP, MN, SR, JH, Data curation, Methodology; EBW, MS, Obtained bronchoalveolar lavage clinical samples containing alveolar macrophages; GS, Data curation, Investigation; ASP, Resources, Methodology; GL, Data curation, Investigation, Methodology, ; AS, Conceptualization, Data curation, Software, Formal analysis, Supervision, Funding acquisition, Validation, Investigation, Visualization, Methodology, Writing—original draft, Project administration, Writing—review and editing

## Author ORCIDs

Alex Sigal, http://orcid.org/0000-0001-8571-2004

## Ethics

Human subjects: Blood was obtained from adult healthy volunteers after written informed consent (University of KwaZulu-Natal Institutional Review Board approval BE022/13). Alveolar macrophages were obtained from bronchoalveolar lavage as part of an indicated diagnostic procedure after written informed consent (University of KwaZulu-Natal Institutional Review Board approval BE037/12).

# Additional files

### Supplementary files

• Source Code 1. Source code 1- Image analysis script.

• Source Code 2. Bootstrap script used *Figure 1—figure supplement 5B*.

• Source Code 3. *Figure 9B* simulation.

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
