## [Decision Letter]

Thank you for submitting your article "Intracellular growth of *Mycobacterium Tuberculosis* after macrophage cell death leads to serial killing of host cells" for consideration by *eLife*. Your article has been reviewed by three peer reviewers, including David G Russell (Reviewer #3), and the evaluation has been overseen by a Reviewing Editor and Wendy Garrett as the Senior Editor.

The reviewers have discussed the reviews with one another and the Reviewing Editor has drafted this decision to help you prepare a revised submission.

Summary:

The paper by Mahamed and colleagues is conceptually simple, that the size of bacterial aggregates internalized by host phagocytes determines the outcome of the interaction, and that if this outcome is host cell death it triggers a rapid burst of bacterial replication. This is an attractive concept that may have biological significance in late stage, localized expansion of bacterial numbers that likely drives the breakdown of bacterial containment and mobilizes the tissue damage required for transmission. It is the accelerated intracellular growth in this setting that is particularly novel. The findings are generally consistent with in vivo observations adding relevance to the study. Other strengths of the study are in the careful and sophisticated design of experiments, elegant quantitative live cell imaging and mathematical modeling of macrophage cell death and *M. Tuberculosis* dynamics. Having said this, the description of the potential mechanisms involved is limited and these are not explored. The authors can improve the figures a bit to help the reader. Finally, the heavy emphasis on what occurs in the granuloma is a bit of a stretch when using a pure macrophage culture since the presence of the other cell types can certainly influence the findings.

Essential revisions:

Further rationale for the parameters for the Mtb dynamics simulations and some type of validation or sensitivity analysis should be done. The doubling time of 565 hours (23-24 days!) in live cells is rather remarkable.

A major drawback of the paper is the lack of mechanism. There is no mechanism for why Mtb grows better in dead macrophages, i.e. is it in the phagosome, is it metabolizing released lipids, is it the local environment providing acquisition of nutrients into a dying leaky cell, or loss of antimicrobial effector mechanisms or others? The discussion is largely a recapitulation of the data. Nor is their definition of the mechanism on how TB induces cell death based on bacilli number and/or clumps. The authors should further consider potential mechanisms for the observations.

Another major drawback is some methodology. A strain of Mtb expressing RFP is used and the amount of TB quantified by measuring the amount of protein. In Figure 7, an increase in TB of 1.1 fold is seen at 3 hours, since the doubling time is about 20 hrs, is this is due to an increase in surface area preceding cell division, or the division of bacteria that were already in that process prior to infection? There is no validation with quantification of actual TB. CFUs should be done on some experiments.

The figures are very confusing. For example, in Figure 1, is this an example and what exactly does the rank mean, not what is on the right? The stats are not clear in the text, figure or legend. I am not convinced that this represents that the aggregate size determines the probability of macrophage death. Figure 2 does not clearly demonstrate what is claimed. It should be obvious. These are two examples that permeate the figures.

Consider dropping the last sentence. The days of "might provide a novel target for therapeutic intervention" without any rationale or route to intervention have long since past.

---

## [Author Response]

*Summary:*

*The paper by Mahamed and colleagues is conceptually simple, that the size of bacterial aggregates internalized by host phagocytes determines the outcome of the interaction, and that if this outcome is host cell death it triggers a rapid burst of bacterial replication. This is an attractive concept that may have biological significance in late stage, localized expansion of bacterial numbers that likely drives the breakdown of bacterial containment and mobilizes the tissue damage required for transmission. It is the accelerated intracellular growth in this setting that is particularly novel. The findings are generally consistent with* in vivo *observations adding relevance to the study. Other strengths of the study are in the careful and sophisticated design of experiments, elegant quantitative live cell imaging and mathematical modeling of macrophage cell death and M. Tuberculosis dynamics. Having said this, the description of the potential mechanisms involved is limited and these are not explored.*

We have now added mechanistic insight from additional experiments and references as described in the response to the Essential revisions below.

*The authors can improve the figures a bit to help the reader.*

We have extensively revised the figures as described in the response to the Essential revisions below.

*Finally, the heavy emphasis on what occurs in the granuloma is a bit of a stretch when using a pure macrophage culture since the presence of the other cell types can certainly influence the findings.*

We have now emphasized that in our macrophage culture system we are looking at a pathogen driven process that could be counteracted by other cell types, particularly T cells. However, the effectiveness of these would be restricted in late stage disease (Discussion section):

“In these experiments we used a pure macrophage culture to investigate the outcome of macrophage-Mtb interactions. […] These mechanisms would counteract the spread of Mtb, but would likely be restricted by Mtb mediated death of the host cell before recognition and targeting by T cells, limited T cell effector function in the granuloma (Egen et al., 2011), lack of T cells in the inner cell layer of granulomas (Ulrichs et al., 2004), and impaired expression of perforin and granulysin observed in Mtb granulomas (Andersson et al., 2007).”

*Essential revisions:*

*Further rationale for the parameters for the Mtb dynamics simulations and some type of validation or sensitivity analysis should be done. The doubling time of 565 hours (23-24 days!) in live cells is rather remarkable.*

The reviewers are correct that 565 hours is too exact a number given the interquartile range of the data, and the fact that at slow growth rates, small changes in slope are heavily magnified in terms of the doubling time, as shown in Figure 10, around a slope of zero:

Author response image 1.**DOI:**
http://dx.doi.org/10.7554/eLife.22028.035

We therefore substitute 565 hours with a doubling time of >100 hours (Results section, subsection “Mtb clumps are not eliminated by macrophage death and rapidly grow inside the dead cell”, third paragraph). This doubling time, corresponding to a slope of ~0.07 (marked by blue vertical line in the figure), is before the doubling time becomes highly sensitive to the slope.

Regarding simulation parameters, a doubling time of one day is what we observed experimentally for Mtb in dead cells. Since the comparison was to a longer doubling time, we chose for simplicity a doubling time of 2 days. This is well within the range of Mtb growth. For example, the doubling time range for extracellular Mtb is 22-75 hours for the data shown in the inset of Figure 5. This is now described in the Results section:

“We compared the number of dead host cells when the Mtb doubling time in dead cells was 1 day, versus when it was 2 days (where two days is within the range of the doubling times observed for extracellular Mtb, which was 22-75 hours for the data presented in the inset of Figure 5).”

To examine the sensitivity of Mtb growth in the system to doubling time, we repeated the stochastic simulations using doubling times ranging from 1 to 3 days, and asked in what fraction of the simulations the number of Mtb expanded by more than 3 orders of magnitude (from 1 to >1000 bacilli).

The result shows that in a system where 1) Mtb growth depends on macrophage death, and 2) macrophage death depends on the clump size internalized, Mtb expansion is very sensitive to the doubling time in the dead infected cells.

These results are presented as Figure 9—figure supplement 1: Sensitivity of Mtb expansion to doubling time in dead cells. They are also described in the Results section: “High Mtb growth was very sensitive to the doubling time (Figure 9—figure supplement 1)”

A description was also added to the Methods:

“To examine the sensitivity of the number of Mtb after 60 days to doubling time, we ranged t_d_ from 1 to 3 days, and calculated the fraction of the simulations in which the number of Mtb expanded by more than 3 orders of magnitude (from 1 to >1000 bacilli).”

*A major drawback of the paper is the lack of mechanism. There is no mechanism for why Mtb grows better in dead macrophages, i.e. is it in the phagosome, is it metabolizing released lipids, is it the local environment providing acquisition of nutrients into a dying leaky cell, or loss of antimicrobial effector mechanisms or others. The discussion is largely a recapitulation of the data.*

To understand the factors allowing for robust Mtb growth in dead but not live macrophages, we performed an experiment where we labelled the Mtb with pHrodo, an Mtb binding dye that fluoresces at low pH. We observed that preceding death, Mtb show pHrodo fluorescence, indicating that in our system Mtb are in an acidified phagosome. Hence, the growth suppression we observe in live cells makes sense. Dramatically, upon cell death, the acidified compartment disappears, as now shown in Video 3 and Figure 6, where the method is shown in Figure 6, and the result summarized in Figure 6.

A description is now added to the Results section:

“To understand the mechanism behind the very slow growth in live MDM versus robust growth in dead MDM, we labelled Mtb with the pH detection dye pHrodo prior to MDM infection. […] Upon cell death, this compartmentalization disappeared.”

The method is described in the Methods section:

“pHrodo labeling. Mtb was labelled with 100 µM pHrodo Green STP Ester (Thermo Fisher) in 100 mM sodium bicarbonate buffer, pH 8.5 for 30 minutes at room temperature, and then washed 3 times with phosphate buffered saline before proceeding to infection of MDMs. The DNA intercalating dye DRAQ7 was used at 3µM in the same experiments to detect cell death.”

This result implies that antimicrobial effector mechanisms were lost upon death. However, this does not fully explain the rapid Mtb growth in dead cells relative to the extracellular medium. If Mtb growth in the dead cells was solely dependent of leakage of nutrients from the extracellular environment, the intracellular bacilli would not be expected to replicate any faster than extracellular Mtb. Instead, our results are consistent with Mtb gaining access to cytosolic nutrient sources, which promote growth in other intracellular pathogens.

We have added a discussion of the mechanisms in the Discussion section:

“[…] the dead cell may provide Mtb with a favorable growth niche relative to the intact phagosome in live cells or the extracellular environment, while live cells provide a poor growth environment either because of antimicrobial effector mechanisms or reduced access to nutrients in the Mtb compartment. […] If Mtb growth in the dead cells was solely dependent of leakage of nutrients from the extracellular environment, the intracellular bacilli would not be expected to replicate any faster than extracellular Mtb. This is clearly not what we observe.”

*Nor is their definition of the mechanism on how TB induces cell death based on bacilli number and/or clumps. The authors should further consider potential mechanisms for the observations.*

We have now added discussion about possible reasons why clumps cause cell death (Discussion section):

“If breach of the phagolysosomal membrane precedes death in Mtb infection, one reason for the increased cytotoxicity of aggregates may include increased membrane breaching (rodriquez-Muela et al., 2015; Sargeant et al., 2014; Boya and Kroemer, 2008). In addition, Mtb induced mitochondrial membrane breakdown (Duan et al., 2002), interference with host plasma membrane repair (Divangahi et al., 2009; Divangahi et al., 2010), and toxin secretion (Sun et al., 2015) would all be expected to scale with Mtb number per macrophage.”

*Another major drawback is some methodology. A strain of Mtb expressing RFP is used and the amount of TB quantified by measuring the amount of protein. In Figure 7, an increase in TB of 1.1 fold is seen at 3 hours, since the doubling time is about 20 hrs, is this is due to an increase in surface area preceding cell division, or the division of bacteria that were already in that process prior to infection?*

In an unsynchronized culture, an increase of fluorescence from 1 to 1.1 would be the result of some bacteria increasing in surface area and others dividing. We have validated that an incremental increase in fluorescence corresponds to a similar increase in CFU (see point 1 in the discussion of the validation below).

*There is no validation with quantification of actual TB. CFUs should be done on some experiments.*

We have now validated fluorescence as a measurement of bacterial number by CFU in two experiments:

1) We have tracked the growth of our fluorescent strain both by fluorescence and CFU over 3 days. We found a tight correspondence between the two measures. The results are presented as Figure 1—figure supplement 2 and described in the Results section:

“To examine whether Mtb fluorescence is a valid measure of Mtb number, we tracked the increase in Mtb by fluorescence versus by colony forming units (CFU) over 3 days of growth. We found a tight correspondence between the two measures (Figure 1—figure supplement 2), with an incremental increase in fluorescence translating to an incremental increase in the number of bacilli as measured by CFU. Hence, fluorescence measurements reflect Mtb numbers and increase in Mtb fluorescence reflects Mtb growth.”

2) We have redone the experiment comparing intracellular Mtb growth in dead cells versus Mtb extracellular growth. This was possible to do in bulk culture since we could increase the dose of Mtb until 99% of macrophages died, and compare it to the same dose of Mtb, but grown extracellularly. We observed a dramatic difference between cell-free growth and intracellular growth in dead cells (Figure 5—figure supplement 1), validating the conclusions from the time-lapse data.

The data are described in the Results section:

“We confirmed these results by CFU, quantifying the number of Mtb after 3 days of growth in dead MDM versus extracellular growth in MDM medium. […] We recovered approximately an order of magnitude more Mtb by CFU after growth in dead cells versus after growth in the MDM extracellular medium (Figure 5—figure supplement 1).”

*The figures are very confusing. For example, in Figure 1, is this an example and what exactly does the rank mean, not what is on the right?*

We have now represented the data in Figure 1 as a standard heat map, with time on the x-axis and mean number of Mtb internalized on the y-axis. The method by which this was done is described in the figure legend. The previous Figure 1 is now Figure 1—figure supplement 4.The result is described in the Results section:

“To compare the death frequencies between cells internalizing different numbers of Mtb over the course of the imaging, we divided the dataset into ten groups of infected cells, where cells internalizing a similar number of Mtb were grouped together and where each group constituted 10% of the total number of infected cells. […] As the number of bacteria internalized increased, there was more cell death and it occurred sooner (Figure 1).”

Unlike the previous representation of Figure 1, which we agree with the referees was difficult to interpret because of the large amount of data, this simplified and smoothened representation makes it clear that there is a threshold for cell death as a function of the number of Mtb phagocytosed. This is now described in the Results section:

“[…] a threshold for Mtb to induce death was clearly visible in Figure 1 at about 50 Mtb internalized, with differences relative to bystander and lightly infected cells becoming highly significant (Figure 1—figure supplement 5).”

*The stats are not clear in the text, figure or legend. I am not convinced that this represents that the aggregate size determines the probability of macrophage death.*

In order to clarify the calculation of significance, we introduce Figure 1—figure supplement 5. In part A, we show another representation of the data – the fraction of dead cells in each group of cells over time as a curve.

In part B, the p-value for the significance of the death frequency at the end of the movie (last point on each of the curves) is determined by bootstrap and presented as a table, where the frequency of each group of cells is compared to each of the groups with a lower mean number of Mtb. Values which are tested for significance using a stringent significance threshold of 0.005, where α=0.05 for single comparisons was adjusted for n=10 comparisons by the Bonferroni method. The higher stringency resulted in some values on the borderline of significance from previous version of the paper falling below the significance threshold this time. The significant results using this threshold are in bold.

We provide the source code for the bootstrap statistical analysis as Source code 2. The bootstrap is now described in the Figure 1—figure supplement 5 legend.

We now also describe these results in the Results section:

“Phagocytosis of small numbers of Mtb, less than approximately 10 bacilli, did not induce significantly more macrophage death than observed in uninfected bystander cells (Figure 1—figure supplement 5). […] The probability of death increased significantly relative to bystanders and cells with less than 10 Mtb when more than 30 Mtb were internalized per cell over the course of the movie, and a threshold for Mtb to induce death was clearly visible in Figure 1 at about 50 Mtb internalized, with differences relative to bystander and lightly infected cells becoming highly significant (Figure 1—figure supplement 5).”

We clarify the programs used to generate the statistical results in the Methods section:

“Statistical significance Where noted, bootstrap (Efron and Tibshirani, 1994) was used to determine statistical significance. […] Statistical analysis not using bootstrap was performed using Graphpad Prism 6 and identified in the figure legends.”

We now provide two references in the Methods section for the statistical technique and the correction for multiple hypotheses:

81) Efron B & Tibshirani RJ (1994) An introduction to the bootstrap (CRC press).

82) Noble WS (2009) How does multiple testing correction work? Nature biotechnology 27(33):1135-1137.

*Figure 2 does not clearly demonstrate what is claimed. It should be obvious. These are two examples that permeate the figures.*

We have now added summary panel A to Figure 2 that shows the data represented as survival curves.We have kept the single cell representations, but modified them to make it clearer that we are comparing multiple small Mtb pickups to one large pickup by both color and size coding the pickups.

We clarify how we chose the two comparison groups in the Results section, and describe the outcomes:

“In order to control for bacterial number, we limited the comparison to cells which internalized an average of approximately 50 Mtb (49.6 ± 18.9 Mtb for macrophages which phagocytosed multiple smaller aggregates, versus 49.4 ± 14.3 for macrophages which phagocytosed one large aggregate), an infection level where approximately one half of infected cells died by the end of our imaging period of 83 hours (Figure 1—figure supplement 5). […] The difference was significant (p=5x10^-4^ by bootstrap). In addition to the summary of the results (Figure 2), we also represent the individual cell histories (Figure 2 for multiple pickups and Figure 2 for single pickups).”

We have also modified the single cell graphs in Figure 3 and Figure 8 to increase clarity. We note that we are trying to describe a complex process: we track individual macrophages that phagocytose different numbers of Mtb in different states at different times, and may or may not die as a result. Hence, there is an inherent limit to how much the data can be simplified without losing information. We state this in the Results section, and further clarify the single cell representations:

“The data is inherently complex as it tracks individual macrophages which phagocytose different numbers of Mtb in different states at different times, and may or may not die as a result. […] If macrophage death occurs, line color changes from green to dark blue at time of death.”

We also further clarify the single cell representation in Figure 3 in the Results section:

“We therefore compared macrophages which internalized single clumps of Mtb during the first half of the imaging period, to increase the measurable time to death, if it occurred. […] In contrast, the time to cell death appeared to be clearly linked to the time of pickup when the number of Mtb in the clump was large (Figure 3).”

Furthermore, we clarify Figure 4 by simplifying the cell panel In Figure 4 by deleting the DRAQ7 only images and only showing the overlays.We the introduce Figure panel B to make it clear how we quantified the results, and panel C to summarize the results.

*Consider dropping the last sentence. The days of "might provide a novel target for therapeutic intervention" without any rationale or route to intervention have long since past.*

We thank the referees for pointing out that the paper could be concluded better. We have now dropped the last sentence and replaced it with the essential message of the paper, paraphrasing some of the summary provided by the referees:

“The process we observe likely reflects a positive feedback loop for Mtb replication in late stage infection, as illustrated by the outcomes of the simplified model in Figure 9. […] This illustrates how, once initiated, Mtb replication can be locally stabilized in the active state.”